# Newton Informed Neural Operator for Solving Nonlinear Partial Differential Equations

**Wenrui Hao**[*]
Department of Mathematics
The Pennsylvania State University, University Park,
State College, PA, USA
wxh64@psu.edu

**Xinliang Liu**
School of Mathematical Sciences,
Shenzhen University, Shenzhen, China
King Abdullah University of Science and Technology, Thuwal, Saudi Arabia
xinliang.liu@kaust.edu.sa

**Yahong Yang**
Department of Mathematics
The Pennsylvania State University, University Park,
State College, PA, USA
yxy5498@psu.edu

## Abstract

Solving nonlinear partial differential equations (PDEs) with multiple solutions is essential in various fields, including physics, biology, and engineering. However, traditional numerical methods, such as finite element and finite difference methods, often face challenges when dealing with nonlinear solvers, particularly in the presence of multiple solutions. These methods can become computationally expensive, especially when relying on solvers like Newton's method, which may struggle with ill-posedness near bifurcation points. In this paper, we propose a novel approach, the Newton Informed Neural Operator, which learns the Newton solver for nonlinear PDEs. Our method integrates traditional numerical techniques with the Newton nonlinear solver, efficiently learning the nonlinear mapping at each iteration. This approach allows us to compute multiple solutions in a single learning process while requiring fewer supervised data points than existing neural network methods.

## 1 Introduction

Neural networks have been extensively applied to solve partial differential equations (PDEs) in various fields, including biology, physics, and materials science [21, 9]. While much of the existing work focuses on PDEs with a unique solution, nonlinear PDEs with multiple solutions pose a significant challenge [39, 10] but are widely encountered in applications such as [1, 3, 31, 36, 14, 13]. In this paper, we aim to solve the following nonlinear PDEs with multiple solutions:

$$\begin{cases} (\mathcal{L}u)(\boldsymbol{x}) = f(u(\boldsymbol{x})), & \boldsymbol{x} \in \Omega, \\ u(\boldsymbol{x}) = 0, & \boldsymbol{x} \in \partial\Omega, \end{cases} \tag{1}$$

---

[*]Corresponding Authors

38th Conference on Neural Information Processing Systems (NeurIPS 2024).

Here, $\Omega$ is the domain of equation, $f(u)$ is a nonlinear function in $\mathbb{R}$, $u : \mathbb{R}^d \to \mathbb{R}$ and $\mathcal{L}$ is a second-order elliptic operator given by $\mathcal{L}u = -\sum_{i,j=1}^d a^{ij}(\boldsymbol{x})u_{x_i x_j} + \sum_{i=1}^d b^i(\boldsymbol{x})u_{x_i} + c(\boldsymbol{x})u$, for given coefficient functions $a^{ij}, b^i, c(i,j = 1, \ldots, d)$ with $\sum_{i,j=1}^n a^{ij}(\boldsymbol{x})\xi_i\xi_j \geq \lambda|\boldsymbol{\xi}|^2$, for a constant $\lambda \geq 0$.

Various neural network methods have been developed to tackle partial differential equations (PDEs), including PINN [33], the Deep Ritz method [45], DeepONet [29], FNO [25], MgNO [16], and OL-DeepONet [26]. These methods can be broadly categorized into two types: function learning and operator learning approaches. In function learning, the goal is to directly learn the solution. However, these methods often encounter the limitation of only being able to learn one solution in each learning process. Furthermore, the problem becomes ill-posed when there are multiple solutions. On the other hand, operator learning aims to approximate the map between parameter functions in PDEs and the unique solution. This approach cannot address the issue of multiple solutions or find them in a single training session. We will discuss this in more detail in the next section.

In this paper, we present a novel neural network approach for solving nonlinear PDEs with multiple solutions. Our method is grounded in operator learning, allowing us to capture multiple solutions within a single training process, thus overcoming the limitations of function learning methods in neural networks. Moreover, we enhance our network architecture by incorporating traditional Newton methods [35, 1], as discussed in the next section. This integration ensures that the problem of operator learning becomes well-defined, as Newton's methods provide well-defined locally, thereby ensuring a robust operator. This approach addresses the challenges associated with directly applying operator networks to such problems. Additionally, we leverage Newton information during training, enabling our method to perform effectively even with limited supervised data points. We introduce our network as the **Newton Informed Neural Operator**. To clarify, we do not design a specific neural structure for the neural operator. The Newton information is not incorporated into the architecture of the neural network but rather into the training process. Specifically, the Newton method is incorporated into the loss function, as detailed in Section 3.3.

As mentioned earlier, our approach combines the classical Newton method, which translates nonlinear PDEs into an iterative process involving solving linear functions at each iteration. One key advantage of our method is that, once the operator is effectively learned, there is no need to solve the linear equation at every iteration. This significantly reduces computation time, especially in complex systems encountered in fields such as material science, biology, and chemistry. Furthermore, once the Newton Informed Neural Operator is well-trained, it can be applied to compute new solutions with appropriate initial guesses, even those not present in the training data. Details of this capability are demonstrated in the numerical example of the Gray-Scott model. Overall, the Newton Informed Neural Operator efficiently solves nonlinear PDEs with multiple solutions by learning the Newton nonlinear solver. It addresses the time-consuming nature of traditional nonlinear solvers and requires fewer supervised data points compared to existing neural network methods. Additionally, it saves time by eliminating the need for repeatedly solving nonlinear systems, as is required in traditional Newton methods. Once the neural operator is learned, it can also compute new solutions beyond those provided in the supervised data.

The following paper is organized as follows: In the next section (Section 2), we will delve into nonlinear PDEs with multiple solutions and discuss related works on solving PDEs using neural network methods. In Section 3, we will review the classical Newton method for solving PDEs and introduce the Newton Informed Neural Operator, which combines neural operators with the Newton method to address nonlinear PDEs with multiple solutions. In this section, we will also analyze the approximation and generalization errors of the Newton Informed Neural Operator. Finally, in Section 4, we present the numerical results of our neural networks for solving nonlinear PDEs. The first example demonstrates that the Newton Informed Neural Operator requires minimal data for training, the second example shows that the speed of the Newton Informed Neural Operator is significantly faster than the traditional Newton method, and the last example highlights that the Newton Informed Neural Operator can discover new solutions not present in the supervised data.

## 2 Backgrounds and Relative Works

### 2.1 Nonlinear PDEs with multiple solutions

Significant mathematical models depicting natural phenomena in biology, physics, and materials science are rooted in nonlinear partial differential equations (PDEs) [5]. These models, characterized by their inherent nonlinearity, present complex multi-solution challenges. Illustrative examples include string theory in physics, reaction-diffusion systems in chemistry, and pattern formation in biology [20, 12]. However, experimental techniques like synchrotronic and laser methods can only observe a subset of these multiple solutions. Thus, there is an urgent need to develop computational methods to unravel these nonlinear models, offering deeper insights into the underlying physics and biology [17]. Consequently, efficient numerical techniques for identifying these solutions are pivotal in understanding these intricate systems. Despite recent advancements in numerical methods for solving nonlinear PDEs, significant computational challenges persist for large-scale systems. Specifically, the computational costs of employing Newton and Newton-like approaches are often prohibitive for the large-scale systems encountered in real-world applications. In response to these challenges [15, 19], we propose an operator learning approach based on Newton's method to efficiently solve nonlinear PDEs.

### 2.2 Related works

Indeed, there are numerous approaches to solving partial differential equations (PDEs) using neural networks. Broadly speaking, these methods can be categorized into two main types: function learning and operator learning.

In function learning, neural networks are used to directly approximate the solutions to PDEs. Function learning approaches aim to directly learn the solution function itself. On the other hand, in operator learning, the focus is on learning the operator that maps input parameters to the solution of the PDE. Instead of directly approximating the solution function, the neural network learns the underlying operator that governs the behavior of the system.

**Function learning methods**    In function learning, a commonly employed method for addressing this problem involves the use of Physics-Informed Neural Network (PINN)-based learning approaches, as introduced by Raissi et al. [33], and Deep Ritz Methods [45]. However, in these methods, the task becomes particularly challenging due to the ill-posed nature of the problem arising from multiple solutions. Despite employing various initial data and training methods, attaining high accuracy in solution learning remains a complex endeavor. Even when a high-accuracy solution is achieved, each learning process typically results in the discovery of only one solution. The specific solution learned by the neural network is heavily influenced by the initial conditions and training methods employed. However, discerning the relationships between these factors and the learned solution remains a daunting task. In [19], the authors introduce HomPINNs for learning multiple solutions to PDEs, where the number of solutions that can be learned depends on the choice of "start functions." However, if the "start functions" are not appropriately selected, HomPINNs may fail to capture all solutions. In this paper, we present an operator learning approach combined with Newton's method to train the nonlinear solver. While this approach is not specifically designed for computing multiple solutions, it can be employed to compute them if suitable initial guesses are provided.

**Operator learning methods**    Various approaches have been developed for operator learning to solve PDEs, including DeepONet [29], which integrates physical information [7, 26], as well as techniques like FNO [25] inspired by spectral methods, and MgNO [16], HANO [27], and WNO [24] based on multilevel methods, and transformer-based neural operators [4, 8]. These methods focus on approximating the operator between the parameters and the solutions. Firstly, they require the solutions of PDEs to be unique; otherwise, the operator is not well-defined. Secondly, they focus on the relationship between the parameter functions and the solution, rather than the initial data and multiple solutions.

# 3 Newton Informed Neural Operator

## 3.1 Review of Newton Methods to Solve Nonlinear Partial Differential Equations

To tackle this problem Eq. (1), we employ Newton's method by linearizing the equation.

For the Newton method applied to an operator, if we aim to find the solution of $\mathcal{F}(u) = 0$, the iteration can be written as:

$$\mathcal{F}'(u_n)u_{n+1} = \mathcal{F}'(u_n)u_n - \mathcal{F}(u_n) \Longleftrightarrow \mathcal{F}'(u_n)\delta u = -\mathcal{F}(u_n),$$

where $\delta u = u_{n+1} - u_n$.

In this context, $\mathcal{F}'(u)v$ is the (Fréchet) derivative of the operator, which is a linear operator to $v$, defined as follows: To find $\mathcal{F}'(u)$ in $\mathcal{X}$, for any $v \in \mathcal{X}$,

$$\lim_{|v| \to 0} \frac{|\mathcal{F}(u+v) - \mathcal{F}(u) - \mathcal{F}'(u)v|}{|v|} = 0,$$

where $|\cdot|$ denotes the norm in $\mathcal{X}$.

For solving Eq. (1), given any initial guess $u_0(\boldsymbol{x})$, for $i = 1, 2, \ldots, M$, in the $i$-th iteration of Newton's method, we have $\tilde{u}(\boldsymbol{x}) = u + \delta u(\boldsymbol{x})$ by solving

$$\begin{cases} (\mathcal{L} - f'(u))\delta u = -\mathcal{L}u + f(u), & \boldsymbol{x} \in \Omega \\ \delta u = 0, & \boldsymbol{x} \in \partial\Omega, \end{cases} \tag{2}$$

which is based on the fact that the (Fréchet) derivative of $\mathcal{L} - f(\cdot)$ at $u$ is $\mathcal{L} - f'(u)$. If Eq. (2) has a unique solution, then by solving Eq. (2) and repeating the process $M$ times, we will obtain one of the solutions of the nonlinear equation (1). Denoting the mapping for $u$ and $\delta u$, the solution of Eq. (2) with parameter $u$, as $\mathcal{G}(u) := \delta u$, we know that

$$\lim_{n \to \infty} (\mathcal{G} + \mathrm{Id})^{(n)}(u_0) = u^*,$$

where $u^*$ is one of the solutions of Eq. (1). For different initial conditions, this process will converge to different solutions of Eq. (1), making this method suitable for finding multiple solutions. Furthermore, the Newton method is well-posed, meaning that each initial condition $u_0$ will converge to a single solution of Eq. (1) under appropriate assumptions (see Assumption 1). This approach helps to address the ill-posedness encountered when using PINNs directly to solve Eq. (1). However, repeatedly solving Eq. (1) can be computationally expensive, especially in high-dimensional cases or when a large number of discrete points are involved. In this paper, we tackle these challenges by employing neural networks.

## 3.2 Neural Operator Structures

In this section, we introduce the structure of the neural operator to approximate the operator locally in the Newton methods from Eq.(2), i.e., $\delta u := \mathcal{G}(u)$, where $\delta u$ is the solution of Eq.(2), which depends on $u$. If we can learn the operator $\mathcal{G}(u)$ well using the neural operator $\mathcal{O}(u; \boldsymbol{\theta})$, then for an initial function $u_0$, assume the $n$-th iteration will approximate one solution, i.e., $(\mathcal{G} + \mathrm{Id})^{(n)}(u_0) \approx u^*$. Thus,

$$(\mathcal{O} + \mathrm{Id})^{(n)}(u_0) \approx (\mathcal{G} + \mathrm{Id})^{(n)}(u_0) \approx u^*.$$

For another initial condition, we can evaluate our neural operator and find the solution directly.

Then we discuss how to train such an operator. To begin, we define the following shallow neural operators with $p$ neurons for operators from $\mathcal{X}$ to $\mathcal{Y}$ as

$$\mathcal{O}(a; \boldsymbol{\theta}) = \sum_{i=1}^{p} \mathcal{A}_i \sigma \left( \mathcal{W}_i a + \mathcal{B}_i \right) \quad \forall a \in \mathcal{X} \tag{3}$$

where $\mathcal{W}_i \in \mathcal{L}(\mathcal{X}, \mathcal{Y}), \mathcal{B}_i \in \mathcal{Y}, \mathcal{A}_i \in \mathcal{L}(\mathcal{Y}, \mathcal{Y})$, and $\boldsymbol{\theta}$ denote all the parameters in $\{\mathcal{W}_i, \mathcal{A}_i, \mathcal{B}_i\}_{i=1}^{p}$. Here, $\mathcal{L}(\mathcal{X}, \mathcal{Y})$ denotes the set of all bounded (continuous) linear operators between $\mathcal{X}$ and $\mathcal{Y}$, and $\sigma : \mapsto \mathbb{R}$ defines the nonlinear point-wise activation.

In this paper, we will use shallow DeepONet [29, 22] to approximate the Newton operator. To provide a more precise description, in the shallow neural network, $\mathcal{W}_i$ represents an interpolation of operators. With proper and reasonable assumptions, we can present the following theorem to ensure that DeepONet can effectively approximate the Newton method operator. The proof will be provided in the appendix. Furthermore, MgNO is replaced by $\mathcal{W}$ as a multigrid operator [38], and FNO is some kind of kernel operator; our analysis can be generalized to such cases.

Before the proof, we need to establish some assumptions regarding the input space $\mathcal{X} \subset H^2(\Omega)$ of the operator and $f(u)$ in Eq. (1). The definition of the Sobolev space can be found in Appendix B.1.

**Assumption 1.** *(i): For any $u \in \mathcal{X}$, we have that the linear equation*

$$(\mathcal{L} - f'(u))\delta u = -\mathcal{L}u + f(u)$$

*is well-posed for solving $\delta u$.*

*(ii): There exists a constant $F$ such that $\|f(x)\|_{W^{2,\infty}(\mathbb{R})} \leq F$.*

*(iii): All coefficients in $\mathcal{L}$ are $C^1$ and $\partial\Omega \in C^2$.*

*(iv): $\mathcal{X}$ has a Schauder basis $\{b_k\}_{k=1}^\infty$, we define the canonical projection operator $\mathcal{P}_n$ based on this basis. The operator $\mathcal{P}_n$ projects any element $u \in \mathcal{X}$ onto the finite-dimensional subspace spanned by the first $n$ basis elements $\{b_1, b_2, \ldots, b_n\}$. Specifically, for $u \in \mathcal{X}$, $u = \sum_{k=0}^\infty \alpha_k b_k$, let $\mathcal{P}_n(u) = \sum_{k=0}^n \alpha_k b_k$, where $\alpha_k$ are the coefficients in the expansion of $u$ with respect to the basis $\{b_n\}$. The canonical projection operator $\mathcal{P}_n$ is a linear bounded operator on $\mathcal{X}$. According to the properties of Schauder bases, these projections $\mathcal{P}_n$ are uniformly bounded by some constant $C$. Furthermore, we assume, for any $u \in \mathcal{X}$, $\epsilon > 0$, there exists a $n$ such that*

$$\|u - \mathcal{P}_n u\|_{H^2(\Omega)} \leq \epsilon, \quad \text{for all } u \in \mathcal{X}.$$

More discussion about the assumption is shown in the appendix. The sketch of the proof is illustrated in Fig. 1.

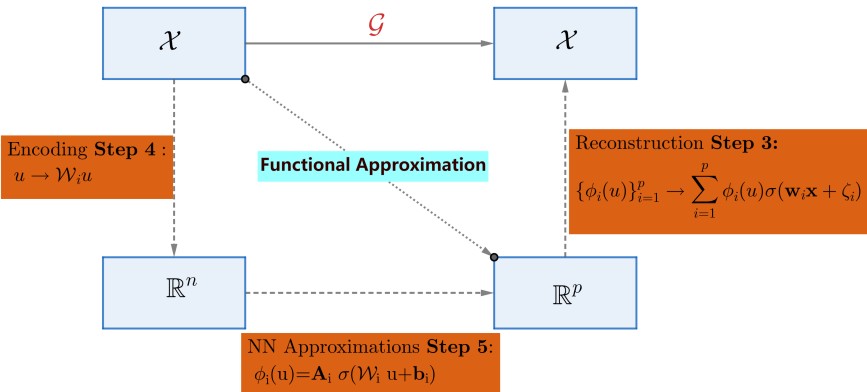

Figure 1: The sketch of proof for Theorem 1.

**Theorem 1.** *Suppose $\mathcal{X} = \mathcal{Y} \subset H^2(\Omega)$ and Assumption 1 holds. Then, there exists a neural network $\mathcal{O}(u; \boldsymbol{\theta}) \in \Xi_p$ defined as*

$$\Xi_p := \left\{ \sum_{i=1}^p \boldsymbol{A}_i \sigma\left(\mathcal{W}_i u + \boldsymbol{b}_i\right) \sigma\left(\boldsymbol{w}_i \cdot \boldsymbol{x} + \zeta_i\right) \Big| \mathcal{W}_i \in \mathcal{L}(\mathcal{X}, \mathbb{R}^m), \boldsymbol{b}_i \in \mathbb{R}^m, \boldsymbol{A}_i \in \mathbb{R}^{1 \times m} \right\} \quad (4)$$

*such that*

$$\sup_{u \in \mathcal{X}} \left\| \sum_{i=1}^p \boldsymbol{A}_i \sigma\left(\mathcal{W}_i u + \boldsymbol{b}_i\right) \sigma\left(\boldsymbol{w}_i \cdot \boldsymbol{x} + \zeta_i\right) - \mathcal{G}(u) \right\|_{L^2(\Omega)} \leq C_1 m^{-\frac{1}{n}} + C_2(\epsilon + p^{-\frac{2}{d}}), \quad (5)$$

*where $\sigma$ is a smooth non-polynomial activation function, $n$ is shown in Assumption 1 and contained in $\mathcal{W}_i$, $C_1$ is a constant independent of $m$, $\epsilon$, and $p$, $C_2$ is a constant depended on $p$, $n$ and $F$ (see in Assumption 1) is the scale of the $\mathcal{P}$ in Assumption 1. And $\epsilon$ depends on $n$. $n$ and $\epsilon$ are defined in Assumption 1.*

The approximation results of DeepONet in Sobolev training can be found in [40].

### 3.3 Loss Functions of Newton Informed Neural Operator

#### 3.3.1 Mean Square Loss

The Mean Square Error loss function is defined as:

$$\mathcal{E}_S(\boldsymbol{\theta}) := \frac{1}{M_u \cdot M_x} \sum_{j=1}^{M_u} \sum_{k=1}^{M_x} |\mathcal{G}(u_j)(\boldsymbol{x}_k) - \mathcal{O}(u_j; \boldsymbol{\theta})(\boldsymbol{x}_k)|^2 \tag{6}$$

where $u_1, u_2, \ldots, u_{M_u} \sim \mu$ are independently and identically distributed (i.i.d) samples in $\mathcal{X}$, and $\boldsymbol{x}_1, \boldsymbol{x}_2, \ldots, \boldsymbol{x}_{M_x}$ are uniformly i.i.d samples in $\Omega$.

However, using only the Mean Squared Error loss function is not sufficient for training to learn the Newton method, especially since in most cases, we do not have enough data $\{u_j, \mathcal{G}(u_j)\}_{j=1}^{M_u}$. Furthermore, there are situations where we do not know how many solutions exist for the nonlinear equation (1). If the data is sparse around one of the solutions, it becomes impossible to effectively learn the Newton method around that solution.

Given that $\mathcal{E}_S(\boldsymbol{\theta})$ can be viewed as the finite data formula of $\mathcal{E}_{Sc}(\boldsymbol{\theta})$, where

$$\mathcal{E}_{Sc}(\boldsymbol{\theta}) = \lim_{M_u, M_x \to \infty} \mathcal{E}_S(\boldsymbol{\theta}).$$

The smallness of $\mathcal{E}_{Sc}$ can be inferred from Theorem 1. To understand the gap between $\mathcal{E}_{Sc}(\boldsymbol{\theta})$ and $\mathcal{E}_S(\boldsymbol{\theta})$, we can rely on the following theorem. Before the proof, we need some assumptions about the data in $\mathcal{E}_S(\boldsymbol{\theta})$:

**Assumption 2.** *(i) Boundedness: For any neural network with bounded parameters, characterized by a bound $B$ and dimension $d_{\boldsymbol{\theta}}$, there exists a function $\Psi : L^2(\Omega) \to [0, \infty)$ such that*

$$|\mathcal{G}(u)(\boldsymbol{x})| \leqslant \Psi(u), \quad \sup_{\boldsymbol{\theta} \in [-B,B]^{d_{\boldsymbol{\theta}}}} |\mathcal{O}(u; \boldsymbol{\theta})(\boldsymbol{x})| \leqslant \Psi(u), \quad \sup_{\boldsymbol{\theta} \in [-B,B]^{d_{\boldsymbol{\theta}}}} |\mathcal{LO}(u; \boldsymbol{\theta})(\boldsymbol{x})| \leqslant \Psi(u)$$

*for all $u \in \mathcal{X}, \boldsymbol{x} \in \Omega$, and there exist constants $C, \kappa > 0$, such that*

$$\Psi(u) \leqslant C(1 + \|u\|_{H^2})^{\kappa}. \tag{7}$$

*(ii) Lipschitz continuity: There exists a function $\Phi : L^2(\Omega) \to [0, \infty)$, such that*

$$|\mathcal{O}(u; \boldsymbol{\theta})(\boldsymbol{x}) - \mathcal{O}(u; \boldsymbol{\theta}')(\boldsymbol{x})| \leqslant \Phi(u) \|\boldsymbol{\theta} - \boldsymbol{\theta}'\|_{\ell^\infty} \tag{8}$$

*for all $u \in \mathcal{X}, \boldsymbol{x} \in \Omega$, and $\Phi(u) \leqslant C(1 + \|u\|_{H^2(\Omega)})^{\kappa}$, for the same constants $C, \kappa > 0$ as in Eq. (7).*

*(iii) Finite measure: There exists $\alpha > 0$, such that*

$$\int_{H^2(\Omega)} e^{\alpha \|u\|_{H^2(\Omega)}^2} \mathrm{d}\mu(u) < \infty.$$

**Theorem 2.** *If Assumption 2 holds, then the generalization error is bounded by*

$$\sup_{\boldsymbol{\theta} \in [-B,B]^{d_{\boldsymbol{\theta}}}} |\mathbb{E}(\mathcal{E}_S(\boldsymbol{\theta}) - \mathcal{E}_{Sc}(\boldsymbol{\theta}))| \leqslant C \left[ \frac{1}{\sqrt{M_u}} \left( 1 + Cd_{\boldsymbol{\theta}} \log(CB\sqrt{M_u})^{2\kappa+1/2} \right) + \frac{d_{\boldsymbol{\theta}} \sqrt{\log M_x}}{\sqrt{M_x}} \right],$$

*where $C$ is a constant independent of $B$, $d_{\boldsymbol{\theta}}$, $M_x$, and $M_u$. The parameter $\kappa$ is specified in (7). Here, $B$ represents the bound of parameters, and $d_{\boldsymbol{\theta}}$ is the number of parameters.*

The proof of Theorem 2 is presented in Appendix B.3.

**Remark 1.** *Assumption 2 is easily satisfied if we consider $\mathcal{X}$ as the local function set around the solution, which is typically the case in Newton's methods. This aligns with our approach and the working region in the approximation part (see Remark 3). The error $\sup_{\boldsymbol{\theta} \in [-B,B]^{d_{\boldsymbol{\theta}}}} |\mathbb{E}(\mathcal{E}_S(\boldsymbol{\theta}) - \mathcal{E}_{Sc}(\boldsymbol{\theta}))|$ suggests that the network can perform well based on the loss function $\mathcal{E}_S(\boldsymbol{\theta})$. The reasoning is as follows: let $\boldsymbol{\theta}_S = \arg\min_{\boldsymbol{\theta} \in [-B,B]^{d_{\boldsymbol{\theta}}}} \mathcal{E}_S(\boldsymbol{\theta})$ and $\boldsymbol{\theta}_{S_c} = \arg\min_{\boldsymbol{\theta} \in [-B,B]^{d_{\boldsymbol{\theta}}}} \mathcal{E}_{Sc}(\boldsymbol{\theta})$. We aim for $\mathbb{E}\mathcal{E}_{Sc}(\boldsymbol{\theta}_S)$ to be small, which can be written as:*

$$\mathbb{E}\mathcal{E}_{Sc}(\boldsymbol{\theta}_S) \leq \mathcal{E}_{Sc}(\boldsymbol{\theta}_{S_c}) + \mathbb{E}(\mathcal{E}_S(\boldsymbol{\theta}_S) - \mathcal{E}_{Sc}(\boldsymbol{\theta}_S)) \leq \mathcal{E}_{Sc}(\boldsymbol{\theta}_{S_c}) + \sup_{\boldsymbol{\theta} \in [-B,B]^{d_{\boldsymbol{\theta}}}} |\mathbb{E}(\mathcal{E}_S(\boldsymbol{\theta}) - \mathcal{E}_{Sc}(\boldsymbol{\theta}))|,$$

*where $\mathcal{E}_{Sc}(\boldsymbol{\theta}_{S_c})$ is small, as demonstrated by Theorem 1 when $B$ is sufficiently large.*

### 3.3.2 Newton Loss

As we have mentioned, relying solely on the MSE loss function can require a significant amount of data to achieve the task. However, obtaining enough data can be challenging, especially when the equation is complex and the dimension of the input space is large. Hence, we need to consider another loss function to aid learning, which is the physical information loss function [33, 7, 19, 24], referred to here as the Network loss function.

The Newton loss function is defined as:

$$\mathcal{E}_N(\boldsymbol{\theta}) := \frac{1}{N_u \cdot N_x} \sum_{j=1}^{N_u} \sum_{k=1}^{N_x} |(\mathcal{L} - f'(u_j))\mathcal{O}(u_j; \boldsymbol{\theta})(\boldsymbol{x}_k) + (\mathcal{L}u_j - f(u_j))(\boldsymbol{x}_k)|^2 \tag{9}$$

where $u_1, u_2, \ldots, u_{N_u} \sim \nu$ are independently and identically distributed (i.i.d) samples in $\mathcal{X}$, and $\boldsymbol{x}_1, \boldsymbol{x}_2, \ldots, \boldsymbol{x}_{N_x}$ are uniformly i.i.d samples in $\Omega$.

Given that $\mathcal{E}_N(\boldsymbol{\theta})$ can be viewed as the finite data formula of $\mathcal{E}_{Nc}(\boldsymbol{\theta})$, where

$$\mathcal{E}_{Nc}(\boldsymbol{\theta}) = \lim_{N_u, N_x \to \infty} \mathcal{E}_S(\boldsymbol{\theta}).$$

To understand the gap between $\mathcal{E}_{Nc}(\boldsymbol{\theta})$ and $\mathcal{E}_N(\boldsymbol{\theta})$, we can rely on the following theorem:

**Corollary 1.** *If Assumption 2 holds, then the generalization error is bounded by*

$$\sup_{\boldsymbol{\theta} \in [-B, B]^{d_{\boldsymbol{\theta}}}} |\mathbb{E}(\mathcal{E}_N(\boldsymbol{\theta}) - \mathcal{E}_{Nc}(\boldsymbol{\theta}))| \leqslant C \left[ \frac{1}{\sqrt{N_u}} \left( 1 + Cd_{\boldsymbol{\theta}} \log(CB\sqrt{N_u})^{2\kappa + 1/2} \right) + \frac{d_{\boldsymbol{\theta}}\sqrt{\log N_x}}{\sqrt{N_x}} \right],$$

*where $C$ is a constant independent of $B$, $d_{\boldsymbol{\theta}}$, $N_x$, and $N_u$. The parameter $\kappa$ is specified in (7). Here, $B$ represents the bound of parameters, and $d_{\boldsymbol{\theta}}$ is the number of parameters.*

The proof of Corollary 1 is similar to that of Theorem 2; therefore, it will be omitted from the paper.

**Remark 2.** *If we only utilize $\mathcal{E}_S(\boldsymbol{\theta})$ as our loss function, as demonstrated in Theorem 2, we require both $M_u$ and $M_x$ to be large, posing a significant challenge when dealing with complex nonlinear equations. Obtaining sufficient data becomes a critical issue in such cases. In this paper, we integrate Newton's information into the loss function, defining it as follows:*

$$\mathcal{E}(\boldsymbol{\theta}) := \lambda \mathcal{E}_S(\boldsymbol{\theta}) + \mathcal{E}_N(\boldsymbol{\theta}), \tag{10}$$

*where $\mathcal{E}_N(\boldsymbol{\theta})$ represents the cost associated with the unsupervised learning data. If we lack sufficient data for $\mathcal{E}_S(\boldsymbol{\theta})$, we can adjust the parameters by selecting a small $\lambda$ and increasing $N_x$ and $N_u$. This strategy enables effective learning even when data for $\mathcal{E}_S(\boldsymbol{\theta})$ is limited. We refer to this neural operator, which incorporates Newton information, as the **Newton Informed Neural Operator**.*

In the following experiment, we will use the neural operator established in Eq. (3) and the loss function in Eq. (10) to learn one step of the Newton method locally, i.e., the map between the input $u$ and the solution $\delta u$ in eq. (2). If we have a large dataset, we can choose a large $\lambda$ in $\mathcal{E}(\boldsymbol{\theta})$ (10); if we have a small dataset, we will use a small $\lambda$ to ensure the generalization of the operator is minimized. After learning one step of the Newton method using the operator neural networks, we can easily and quickly obtain the solution by the initial condition of the nonlinear PDEs (1) and find new solutions not present in the datasets.

## 4 Experiments

### 4.1 Experimental Settings

We introduce two distinct training methodologies. The first approach employs exclusively supervised data, leveraging the Mean Squared Error Loss (6) as the primary optimization criterion. The second method combines both supervised and unsupervised learning paradigms, utilizing a hybrid loss function 10 that integrates Mean Squared Error Loss (6) for small proportion of data with ground truth (supervised training dataset) and with Newton's loss (9) for large proportion of data without ground truth (unsupervised training dataset). We call the two methods **method 1** and **method 2**. The approaches are designed to simulate a practical scenario with limited data availability, facilitating a comparison between these training strategies to evaluate their efficacy in small supervised data regimes. We chose the same configuration of the neural operator (DeepONet) which is aligned with our theoretical analysis. One can find the detailed experimental settings and the datasets for each example below in Appendix A.

## 4.2 Case 1: Convex problem

We consider 2D convex problem $\mathcal{L}(u) - f(u) = 0$, where $\mathcal{L}(u) := -\Delta u$, $f(u) : -u^2 + \sin 5\pi(x+y)$ and $u = 0$ on $\partial\Omega$. We investigate the training dynamics and testing performance of neural operator (DeepONet) trained with two methods, focusing on Mean Squared Error (MSE) and Newton's loss functions. For method 1, we use 500 supervised data samples (with ground truth), while for method 2, we use 5000 unsupervised data samples (only with the initial state) along with supervised data samples, employing the regularized loss function as defined in Equation 10 with $\lambda = 0.01$. Please refer A.1.1 for the dataset generation of convex case. For the detailed experimental settings, refer to Appendix A.

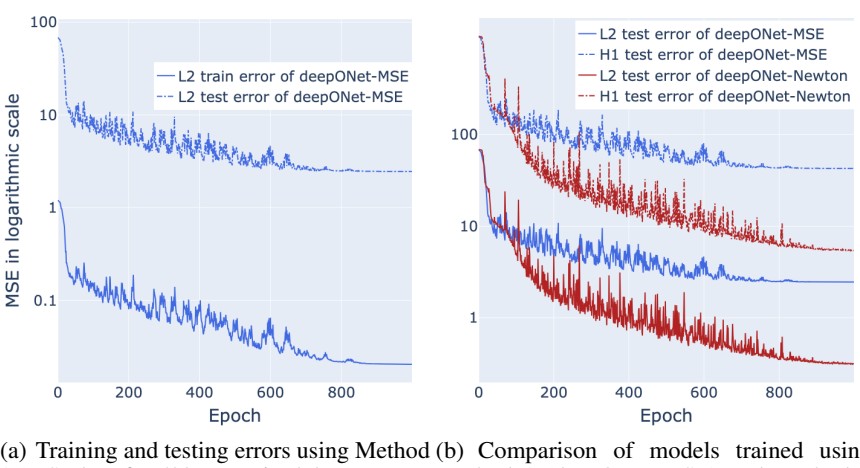

(a) Training and testing errors using Method 1 (MSE loss for 500 supervised data)

(b) Comparison of models trained using Method 1 (deepONet-MSE) and Method 2 (deepONet-Newton)

Figure 2: Training and testing performance of DeepONet under different conditions.

**MSE Loss Training (Fig. 2(a))**: In method 1, Effective training is observed but exhibits poor generalization. The significantly larger testing error compared to the training error suggests that using only MSE loss is insufficient. **Performance Comparison (Fig. 2(b))**: DeepONet-Newton model (Method 2) exhibits superior performance in both $L_2$ and $H_1$ error metrics, highlighting its enhanced generalization accuracy. This study shows the advantages of using Newton's loss for training DeepONet models, illustrating that increasing the number of unsupervised samples via introducing Newton's loss leads to a substantial improvement in the test $L_2$ error and $H_1$ error.

## 4.3 Case 2: Non-convex problem with multiple solutions

We consider a 2D Non-convex problem,

$$\begin{cases} -\Delta u(x,y) - u^2(x,y) = -s\sin(\pi x)\sin(\pi y) & \text{in} \quad \Omega, \\ u(x,y) = 0, & \text{in} \quad \partial\Omega \end{cases} \tag{11}$$

where $\Omega = (0,1) \times (0,1)$ [3]. In this case, $\mathcal{L}(u) := -\Delta(u)$, $f(u) := u^2 - s\sin(\pi x)\sin(\pi y)$ and it has multiple solutions (see Figure 3 for its solutions).

In the experiment, we let one of the multiple ground truth solutions rarely touched in the supervised training dataset such that the neural operator trained via **method 1** will saturate in terms of test error because it relies on the ground truth to recover all the patterns for multiple solution cases (as shown by the curves in Figure 3). On the other hand, the model trained via **method 2** is less affected by the limited supervised data since the utilization of Newton's loss. One can refer to Appendix A for the detailed experiment setting.

**Efficiency** This case study highlights the superior efficiency of our neural operator-based method as a surrogate model for Newton's method. Both methods parallelize operations to solve 500/5000 Newton linear systems simultaneously, each with distinct initial states. The key advantage of the

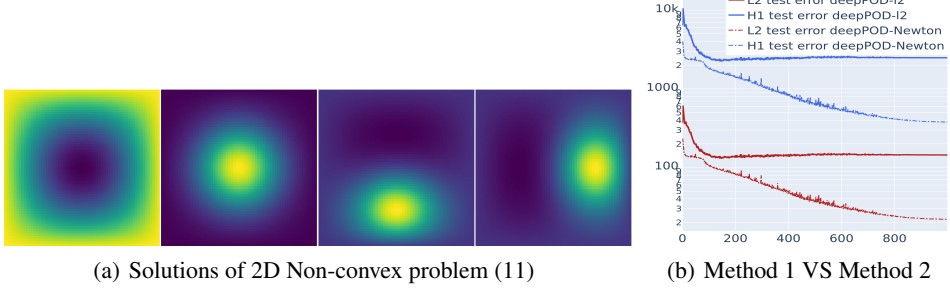

(a) Solutions of 2D Non-convex problem (11)    (b) Method 1 VS Method 2

Figure 3: Solutions of 2D Non-convex problem (11)

neural operator lies in its ability to batch the computation of these independent systems efficiently. By efficiently batching and sampling a wide variety of initial states, the neural operator improves the likelihood of discovering to multiple solutions, particularly in nonlinear PDEs with complex solution landscapes. Consequently, while Newton's method alone does not inherently guarantee finding multiple solutions, the combination of rapid computation and extensive initial condition sampling enhances the chances of identifying multiple solutions.

For a fair comparison, the classical Newton solver was also parallelized using CUDA on a GPU. However, the neural operator naturally handles large batch sizes during inference, allowing it to process all systems in one go. One can find the detailed description of the experiments in A.6.

| Parameter | Newton's Method | NINO |
|---|---|---|
| Number of Streams | 10 | - |
| Data Type | float32 | float32 |
| Execution Time for 500 linear Newton systems (s) | 31.52 | 1.1E-4 |
| Execution Time for 5000 linear Newton systems (s) | 321.15 | 1.4E-4 |

Table 1: Benchmarking the efficiency of Newton Informed Neural Operator. Computational Time Comparison for Solving 500 and 5000 Initial Conditions.

The table demonstrates the significant efficiency gain achieved by batching the computation of independent Newton systems with distinct initial states using the neural operator. For NINO, solving 5000 independent Newton linear systems scales up minimally compared to solving 500 systems, while the classical solver experiences a tenfold increase in computation time. This efficient batching is crucial for improving performance, particularly in complex nonlinear systems like the Gray-Scott model, where solving numerous systems simultaneously is essential for effective pattern discovery.

### 4.4 Case 3: The Gray-Scott model

The Gray-Scott model [31, 11] describes the reaction and diffusion of two chemical species, $A$ and $S$, governed by the following equations:

$$\frac{\partial A}{\partial t} = D_A \Delta A - SA^2 + (\mu + \rho)A,$$
$$\frac{\partial S}{\partial t} = D_S \Delta S + SA^2 - \rho(1 - S),$$

where $D_A$ and $D_S$ are the diffusion coefficients, and $\mu$ and $\rho$ are rate constants.

**Newton's Method for Steady-State Solutions**    Newton's method is employed to find steady-state solutions ($\frac{\partial A}{\partial t} = 0$ and $\frac{\partial S}{\partial t} = 0$) by solving the nonlinear system:

$$0 = D_A \Delta A - SA^2 + (\mu + \rho)A,$$
$$0 = D_S \Delta S + SA^2 - \rho(1 - S). \tag{12}$$

The Gray-Scott model is highly sensitive to initial conditions, where even minor perturbations can lead to vastly different emergent patterns. Please refer to Figure 5 for some examples of the patterns. This sensitivity reflects the model's complex, non-linear dynamics that can evolve into a multitude of possible steady states based on the initial setup. Consequently, training a neural operator to map initial conditions directly to their respective steady states presents significant challenges. Such a model must learn from a vast functional space, capturing the underlying dynamics that dictate the transition from any given initial state to its final pattern. This complexity and diversity of potential outcomes is the inherent difficulty in training neural operators effectively for systems as complex as the Gray-Scott model. One can refer to A.1.2 for a detailed discussion on the Gray-Scott model. We employ a Neural Operator as a substitute for the Newton solver in the Gray-Scott model, which recurrently maps the initial state to the steady state.

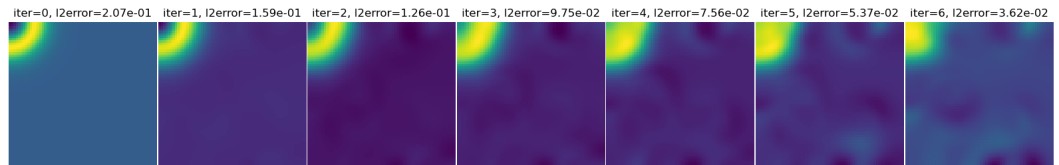

(a) An example demonstrating how the neural operator maps the initial state to the steady state in a iterative manner

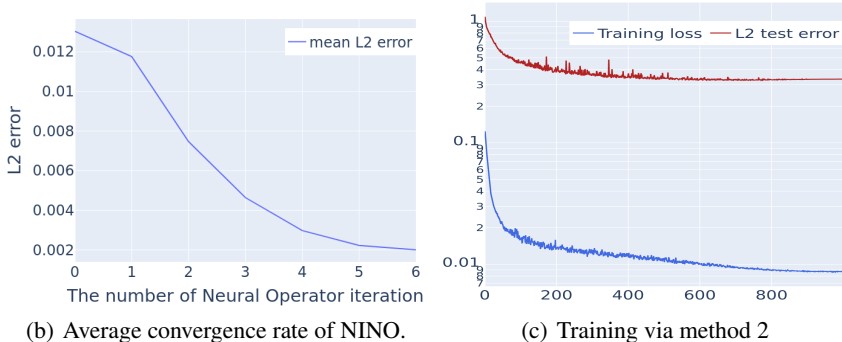

(b) Average convergence rate of NINO.

(c) Training via method 2

Figure 4: The convergence behavior of the Neural Operator-based solver.

In subfigure (a), we use a ring-like pattern as the initial state to test our learned neural operator. This pattern does not appear in the supervised training dataset and lacks corresponding ground truth data. Instead, it is present only in the unsupervised data (Newton's loss), i.e., some data in Newton's loss will converge to this specific pattern. Despite this, our neural operator, trained using Newton's loss, can effectively approximate the mapping of the initial solution to its correct steady state. we further test our neural operator, utilizing it as a surrogate for Newton's method to address nonlinear problems with an initial state drawn from the test dataset. The curve shows the average convergence rate of $\|u - u_i\|$ across the test dataset, where $u_i$ represents the prediction at the $i$-th step by the neural operator. In subfigure (c), we compare the Training Loss (Rescaled Newton's Loss) and Absolute L2 Test Error. The magnitudes are not directly comparable as they represent different metrics; however, the trends are consistent, indicating that the inclusion of unsupervised data and training with Newton's loss contributes to improved model performance.

## 5 Conclusion

In this paper, we develop neural operators to learn the Newton's solver related to nonlinear PDEs (Eq. (1)) with multiple solutions. To speed up the computation of multiple solutions for nonlinear PDEs, we combine neural operator learning with classical Newton methods, resulting in the Newton-informed neural operator. We provide a theoretical analysis of our neural operator, demonstrating that it can effectively learn the Newton operator, reduce the number of required supervised data, and learn solutions not present in the supervised learning data due to the addition of the Newton loss (9) in the loss function. Our experiments are consistent with our theoretical analysis, showcasing the advantages of our network as mentioned earlier.

**Reproducibility Statement**  Code Availability: The code used in our experiments can be accessed via `https://github.com/xlliu2017/learn_newton` and also the supplementary material. Datasets can be downloaded via URLs in the repository. This encompasses all scripts, functions, and auxiliary files necessary to reproduce our results. Configuration Transparency: All configurations, including hyperparameters, model architectures, and optimization settings, are explicitly provided in the Appendix.

## Acknowledgments and Disclosure of Funding

Y.Y. and W.H. were supported by the National Institute of General Medical Sciences through grant 1R35GM146894. The work of X.L. was partially supported by the KAUST Baseline Research Fund.

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

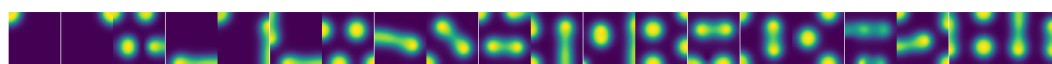

Figure 5: Examples of steady states of the Gray Scott model

# A  Experimental settings

## A.1  Background on the PDEs and generation of datasets

### A.1.1  Case 1: convex problem

**Function and Jacobian**  The function $F(u)$ might typically be defined as:

$$F(u) = -\Delta u + u^2$$

The Jacobian $J(u)$, for the given function $F(u)$, involves the derivative of $F$ with respect to $u$, which includes the Laplace operator and the derivative of the nonlinear term:

$$J(u) = -\Delta + 2 \cdot u.$$

The dataset are generated by sampling the initial state $u_0 \sim \mathcal{N}(0, \Delta^{-3})$ and then calculate the convergent sequence $\{u_0, u_1, ..., u_n\}$ by Newton's method. Each convergent sequence $\{u_0, u_1, ..., u_n\}$ is one data entry in the dataset.

The analysis of function and Jacobian for the non-convex problem (case 2) is similar to the convex problem except that its Jacobian $J(u) = \Delta - 2u$ such that Newton's system is not positive definite.

### A.1.2  Gray Scott model

**Jacobian Matrix**  The Jacobian matrix $J$ of the system is crucial for applying Newton's method:

$$J = \begin{bmatrix} J_{AA} & J_{AS} \\ J_{SA} & J_{SS} \end{bmatrix}$$

with components:

$$\begin{aligned}
J_{AA} &= -D_A \Delta + \text{diag}(-2SA + \mu + \rho), \\
J_{AS} &= \text{diag}(-A^2), \\
J_{SA} &= \text{diag}(2SA), \\
J_{SS} &= -D_S \Delta + \text{diag}(A^2 + \rho).
\end{aligned}$$

The numerical simulation of the Gray-Scott model was configured with the following parameters:

- **Grid Size**: The simulation grid is square with $N = 63$ points on each side, leading to a total of $N^2$ grid points. This resolution was chosen to balance computational efficiency with spatial resolution sufficient to capture detailed patterns. The spacing between each grid point, $h$, is computed as $h = \frac{1.0}{N-1}$. This ensures that the domain is normalized to a unit square, which simplifies the analysis and scaling of diffusion rates.

- **Diffusion Coefficients**: The diffusion coefficients for species $A$ and $S$ are set to $D_A = 2.5 \times 10^{-4}$ and $D_S = 5.0 \times 10^{-4}$, respectively. These values determine the rate at which each species diffuses through the spatial domain.

- **Reaction Rates**: The reaction rate $\mu$ and feed rate $\rho$ are crucial parameters that govern the dynamics of the system. For this simulation, $\mu$ is set to 0.065 and $\rho$ to 0.04, influencing the production and removal rates of the chemical species.

**Simulations**  The simulation utilizes a finite difference method for spatial discretization and Newton's method to solve the steady-state given the initial state. The algorithm is detailed in A.6.

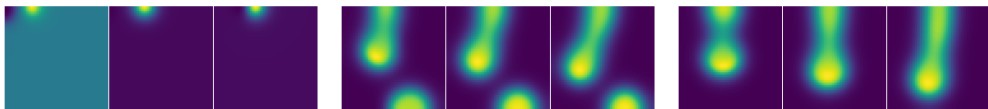

Figure 6: Three examples depicting the evolution from the initial state to the steady state via Newton's method.

## A.2 Data generation

Here is how we generate the supervised data samples:

1. **Step 1:** We use a classical numerical solver to obtain single (multiple) solutions of nonlinear PDEs. For example, in **Case 2**, there exist four solutions $u^1, u^2, u^3, u^4$. We have one solution for **Case 1** and 10 solutions for the **Case 3: Gray-Scott model**.

2. **Step 2:** The supervised dataset is generated by sampling a perturbation around the solution $u^i \sim \mathcal{N}(0, (-\Delta)^{-3})$ on the chosen true solution $u^i$. We then set $u_0^i = u_p^i + u^i$ and calculate the convergent sequence $u_0^i, u_1^i, \ldots, u_n^i$ using Newton's method, which follows the formula:

$$u_{k+1}^i = u_k^i - J_f(u_k^i)^{-1} f(u_k^i).$$

Each convergent sequence $u_0^i, u_1^i, \ldots, u_n^i$ constitutes one supervised data entry in the dataset. In this case, we consider the initial conditions as perturbed, with all perturbations applied around the true solution. A comparison between the traditional method and our proposed method is summarized in Table 1.

For the unsupervised data samples, we only sample the perturbed initial states.

## A.3 Implementations of loss functions

**Discrete Newton's Loss** In solving partial differential equations (PDEs) numerically on a regular grid, the Laplace operator and other differential terms can be efficiently computed using convolution. Here, we detail the method for calculating $J(u)\delta u - F(u)$ where $J(u)$ is the Jacobian matrix, $\delta u$ is the Newton step, and $F(u)$ is the function defining the PDE.

**Discretization** Consider a discrete representation of a function $u$ on a $N \times N$ grid. The function $u$ and its perturbation $\delta u$ are represented as matrices:

$$u, \delta u \in \mathbb{R}^{N \times N}$$

The function $F(u)$, which involves both linear and nonlinear terms, is similarly represented as $F(u) \in \mathbb{R}^{N \times N}$.

**Laplace Operator** Regarding the representing the $J(u)$ with $N \times N$ grid function $u$, the discretized Laplace operator using a finite difference method can be expressed as a convolution:

$$-\Delta u = \begin{bmatrix} 0 & -1 & 0 \\ -1 & 4 & -1 \\ 0 & -1 & 0 \end{bmatrix} * u$$

This convolution computes the result of the Laplace operator applied to the grid function $u$. The boundary conditions can be further incorporated into the convolution with different padding modes. Dirichlet boundary condition corresponds to zeros padding while Neumann boundary condition corresponds to replicate padding.

## A.4 Architecture of DeepONet

A variant of `DeepONet` is used in our Newton-informed neural operator. In the DeepONet, we introduce a hybrid architecture that combines convolutional layers with a static trunk basis, optimized for grid-based data inputs common in computational applications like computational biology and materials science.

**Branch Network**   The branch network is designed to effectively downsample and process the spatial features through a series of convolutional layers:

- A **Conv2D layer** with 128 filters (7x7, stride 2) initiates the feature extraction, reducing the input dimensionality while capturing coarse spatial features.

- This is followed by additional **Conv2D layers** (128 filters, 5x5 kernel, stride 2 and subsequently 3x3 with padding, 1x1) which further refine and compact the feature representation.

- The convolutional output is flattened and processed through two **fully connected layers** (256 units then down to branch features), using GELU activation.

**Trunk Network**   The trunk utilizes a static basis represented by the tensor `V`, incorporated as a non-trainable component: The tensor `V` is precomputed, using Proper Orthogonal Decomposition (POD) as in [28], and is dimensionally compatible with the output of the branch network.

**Forward Pass**   During the forward computation, the branch network outputs are projected onto the trunk's static basis via matrix multiplication, resulting in a feature matrix that is reshaped into the grid dimensionality for output.

### Hyperparameters

The following table 2 summarizes the key hyperparameters used in the `DeepONet` architecture:

| Parameter | Value |
|---|---|
| Number of Conv2D layers | 5 |
| Filters in Conv2D layers | 128, 128, 128, 128, 256 |
| Kernel sizes in Conv2D layers | 7x7, 5x5, 3x3, 1x1, 5x5 |
| Strides in Conv2D layers | 2, 2, 1, 1, 2 |
| Fully Connected Layer Sizes | 256, branch features |
| Activation Function | GELU |

Table 2: Hyperparameters of the DeepONet architecture

## A.5 Training settings

Below we summarize the key configurations and parameters employed in the training for three cases:

### Dataset

- **Case 1**: For method 1, we use 500 supervised data samples (with ground truth) while for method 2, we use 5000 unsupervised data samples (only with the initial state) and 500 supervised data samples.

- **Case 2**: For method 1, we use 5000 supervised data while for method 2, we use 5000 unsupervised data samples and 5000 supervised data samples.

- **Case 3 (Gray-Scott model)**: We only perform method 2, with 10000 supervised data samples and 50000 unsupervised data samples.

### Optimization Technique

- **Optimizer**: Adam, with a learning rate of $1 \times 10^{-4}$ and a weight decay of $1 \times 10^{-6}$.

- **Training Epochs**: The model was trained over 1000 epochs to ensure convergence and we use **Batch Size**: 50.

These settings underscore our commitment to precision and detailed examination of neural operator efficiency in computational tasks. Our architecture and optimization choices are particularly tailored to explore and exploit the capabilities of neural networks in processing complex systems simulations.

### A.6 Benchmarking Newton's Method and neural operator based method

**Experimental Setup**    The benchmark study was conducted to evaluate the performance of a GPU-accelerated implementation of Newton's method, designed to solve systems of linear equations derived from discretizing partial differential equations. The implementation utilized CuPy with CUDA to leverage the parallel processing capabilities of the NVIDIA A100 GPU. The hardware comprises an Intel Cascade Lake 2.5 GHz CPU, and an NVIDIA A100 GPU.

The performance was assessed in terms of total execution time, which includes the setup of matrices and vectors, computation on the GPU, and synchronization of CUDA streams. Both methods leverage the parallel processing capabilities of the GPU. Specifically, the Newton solver explicitly uses 10 streams and CuPy with CUDA to parallelize the computation and fully utilize the GPU parallel processing capabilities, aiming to optimize execution efficiency. In contrast, the neural operator method is inherently parallelized, taking full advantage of the GPU architecture without the explicit use of multiple streams as indicated in the table. The computational times of both methods were evaluated under a common hardware configuration.

**Software Environment:**    Ubuntu 20.04 LTS. **Python Version:** 3.8. **CUDA Version:** 11.4.

Newton's method was implemented to solve the Laplacian equation over a discretized domain. Multiple system solutions were computed in parallel using CUDA streams. The key parameters of the experiment are as follows: **Data Type (dtype):** Single precision floating point (float32). **Number of Streams:** 10 CUDA streams to process data in parallel. **Number of Repeated Calculations:** The Newton method was executed multiple times for 500/5000 Newton linear systems, respectively, distributed evenly across the streams. **Function to Solve Systems:** The CuPy's spsolve was used for solving the sparse matrix systems. The following algorithm A.6 summarizes the procedure to benchmark the time used for solving Newton's system for 5000 different initial states.

---

**Algorithm 1** Solve Newton's Systems on GPU

---

1: **procedure** SOLVESYSTEMSGPU($A$, $u$, $rhs\_f$, $N$)
                           ▷ Precompute RHS and diagonal for all systems
2:   $rhs\_list \leftarrow rhs\_f + u^2 - A \times u$
3:   $diag\_list \leftarrow -2 \times u$
                                 ▷ Initialize solution storage
4:   $delta\_u \leftarrow$ initialize zero matrix with shape of $u^T$
                                     ▷ Solve each system
5:   **for** $i = 0$ **to** $num\_sys - 1$ **do**
6:    $rhs \leftarrow rhs\_list[i]$
7:    $diag \leftarrow diag\_list[i]$
8:    $J \leftarrow A +$ diagonal matrix with $diag$ on the main diagonal
                                 ▷ Solve the linear system
9:    $delta\_u[i] \leftarrow$ spsolve($J, rhs$)
10:   **end for**
11:   **return** transpose($delta\_u$)
12: **end procedure**

---

# B  Supplemental material for proof

## B.1  Preliminaries

**Definition 1** (Sobolev Spaces [6])**.** *Let $\Omega$ be $[0,1]^d$ and let $D$ be the operator of the weak derivative of a single variable function and $D^{\boldsymbol{\alpha}} = D_1^{\alpha_1} D_2^{\alpha_2} \ldots D_d^{\alpha_d}$ be the partial derivative where $\boldsymbol{\alpha} =$*

$[\alpha_1, \alpha_2, \ldots, \alpha_d]^T$ and $D_i$ is the derivative in the $i$-th variable. Let $n \in \mathbb{N}$ and $1 \leq p \leq \infty$. Then we define Sobolev spaces

$$W^{n,p}(\Omega) := \left\{ f \in L^p(\Omega) : D^{\boldsymbol{\alpha}} f \in L^p(\Omega) \text{ for all } \boldsymbol{\alpha} \in \mathbb{N}^d \text{ with } |\boldsymbol{\alpha}| \leq n \right\}$$

with a norm

$$\|f\|_{W^{n,p}(\Omega)} := \left( \sum_{0 \leq |\alpha| \leq n} \|D^\alpha f\|_{L^p(\Omega)}^p \right)^{1/p}$$

if $p < \infty$, and $\|f\|_{W^{n,\infty}(\Omega)} := \max_{0 \leq |\alpha| \leq n} \|D^\alpha f\|_{L^\infty(\Omega)}$.

Furthermore, for $\boldsymbol{f} = (f_1, \ldots, f_d)$, $\boldsymbol{f} \in W^{1,\infty}(\Omega, \mathbb{R}^d)$ if and only if $f_i \in W^{1,\infty}(\Omega)$ for each $i = 1, 2, \ldots, d$ and

$$\|\boldsymbol{f}\|_{W^{1,\infty}(\Omega, \mathbb{R}^d)} := \max_{i=1,\ldots,d} \{\|f_i\|_{W^{1,\infty}(\Omega)}\}.$$

When $p = 2$, denote $W^{n,2}(\Omega)$ as $H^n(\Omega)$ for $n \in \mathbb{N}_+$.

**Proposition 1** ([30]). *Suppose $\sigma$ is a is a continuous non-polynomial function and $K$ is a compact in $\mathbb{R}^d$, then there are positive integers $p$, constants $w_k, \zeta_k$ for $k = 1, \ldots, p$ and bounded linear functionals $c_k : H^r(K) \to \mathbb{R}$ such that for any $v \in H^r(K)$,*

$$\left\| v - \sum_{k=1}^p c_k(v) \sigma \left( \boldsymbol{w}_k \cdot \boldsymbol{x} + \zeta_k \right) \right\|_{L^2(K)} \leq c p^{-r/d} \|v\|_{H^r(K)}. \tag{13}$$

**Proposition 2** ([32, 44]). *Suppose $\sigma$ is a continuous non-polynomial function and $\Omega$ is a compact subset of $\mathbb{R}^d$. For any Lipschitz-continuous function $f$, there exists a shallow neural network such that*

$$\left\| f - \sum_{j=1}^m a_j \sigma \left( \boldsymbol{\omega}_j \cdot \boldsymbol{x} + b_j \right) \right\|_\infty \leq C m^{-1/d}, \tag{14}$$

*where $C$ depends on the Lipschitz constant but is independent of $m$.*

**Lemma 1** ([22]). *The $\epsilon$-covering number of $[-B, B]^d$, $K(\epsilon)$, satisfies*

$$K(\epsilon) \leqslant \left( \frac{CB}{\epsilon} \right)^d,$$

*for some constant $C > 0$, independent of $\epsilon$, $B$, and $d$.*

**Step 5:** Now we estimate

### B.2 Proof of Theorem 1

In this subsection, we present the proof of Theorem 1, which describes the approximation ability of DeepONet.

*Proof of Theorem 1.* **Step 1:** Firstly, we need to verify that the target operator $\mathcal{G}(u)$ is well-defined.

Due to Assumption 1 (i), we know that for $u \in \mathcal{X} \subset H^2(\Omega)$, Equation (2) will have unique solutions. This means that $\mathcal{G}(u)$ is a well-defined operator for the input space $u \in \mathcal{X}$.

**Step 2:** Secondly, we aim to verify that $\mathcal{G}(u)$ is a Lipschitz-continuous operator in $H^2(\Omega)$ for $u \in \mathcal{X}$.

Consider the following:

$$\begin{aligned} f'(u + v) &= f'(u) + v f''(\xi_1) \\ f(u + v) &= f(u) + v f'(\xi_2) \\ \delta_v u(\boldsymbol{x}) &= \delta u(\boldsymbol{x}) + \epsilon(\boldsymbol{x}) \end{aligned} \tag{15}$$

where $\delta u(\boldsymbol{x})$ is the solution of Eq.(2) for the input $u$, and $\delta_v u(\boldsymbol{x})$ is the solution of Eq.(2) for the input $u + v$. Denote

$$\delta_v u(\boldsymbol{x}) - \delta u(\boldsymbol{x}) =: \epsilon(\boldsymbol{x}).$$

Therefore, we have:

$$\begin{cases} (\mathcal{L} - f'(u+v))\epsilon(\boldsymbol{x}) = \Delta v - v(f'(\xi_2) + f''(\xi_1)\delta u), & \boldsymbol{x} \in \Omega \\ \epsilon(\boldsymbol{x}) = 0, & \boldsymbol{x} \in \partial\Omega. \end{cases} \tag{16}$$

Since $u$ and $v$ are in $H^2$ and $\partial\Omega$ is in $C^2$ (Assumption 1 (iii)), according to [6, Theorem 4 in Section 6.3], there exist constants $C$ [2] and $\bar{C}$ such that:

$$\begin{aligned} \|\epsilon(\boldsymbol{x})\|_{H^2(\Omega)} &\leq C\|\mathcal{L}v - v(f'(\xi_2) + f''(\xi_1)\delta u)\|_{L^2(\Omega)} \\ &\leq \bar{C}\|v(\boldsymbol{x})\|_{H^2(\Omega)}. \end{aligned} \tag{17}$$

The last inequality is due to the boundedness of $f'(\xi_2) + f''(\xi_1)\delta u$ (Assumption 1 (ii)).

**Step 3:** In the approximation, we first reduce the operator learning to functional learning.

When the input function $u(\boldsymbol{x})$ belongs to $\mathcal{X} \subset H^2$, the output function $\delta u$ also belongs to $H^2$, provided that $\partial\Omega$ is of class $C^2$. The function $\mathcal{G}(u) = \delta u$ can be approximated by a two-layer network architected by the activation function $\sigma(x)$, which is not a polynomial, in the following form by Proposition 1 [30] (given in Subsection 16):

$$\left\| \mathcal{G}(u)(\boldsymbol{x}) - \sum_{k=1}^{p} c_k[\mathcal{G}(u)]\sigma\left(\boldsymbol{w}_k \cdot \boldsymbol{x} + \zeta_k\right) \right\|_{L^2(\Omega)} \leq C_1 p^{-\frac{2}{d}} \|\mathcal{G}(u)\|_{H^2(\Omega)}, \tag{18}$$

where $\boldsymbol{w}_k \in \mathbb{R}^d$, $\zeta_k \in \mathbb{R}$ for $k = 1, \ldots, p$, $c_k$ is a continuous functional, and $C_1$ is a constant independent of the parameters.

Denote $\phi_k(u) = c_k[\mathcal{G}(u)]$, which is a Lipschitz-continuous functional from $H^2(\Omega)$ to $\mathbb{R}$, which is due to $\mathcal{G}$ is a Lipschitz-continuous operator and $c_k$ is a linear functional. The remaining task in approximation is to approximate these functionals by neural networks.

**Step 4:** In this step, we reduce the functional learning to function learning by applying the operator $\mathcal{P}$ as in Assumption 1 (iv).

Based on $\phi_k(u)$ being a Lipschitz-continuous functional in $H^2(\Omega)$, we have

$$|\phi_k(u) - \phi_k(\mathcal{P}u)| \leq L_k \|u - \mathcal{P}u\|_{H^2(\Omega)} \leq L_k \epsilon,$$

where $L_k$ is the Lipschitz constant of $\phi_k(u)$ for $u \in \mathcal{X}$.

Furthermore, since $\mathcal{P}u$ is an $n$-dimensional term, i.e., it can be denoted by the $n$-dimensional vector $\bar{\mathcal{P}}u \in \mathbb{R}^n$, we can rewrite $\phi_k(\mathcal{P}u)$ as $\psi_k(\bar{\mathcal{P}}u)$, where $\psi_k : \mathbb{R}^n \to \mathbb{R}$ for $l = 1, \ldots, p$. Furthermore, $\psi_k$ is a Lipschitz-continuous function since $\phi_k$ is Lipschitz-continuous and $\mathcal{P}$ is a continuous linear operator.

**Step 5:** In this step, we will approximate $\psi_k$ using shallow neural networks.

Due to Proposition 2, we have that there is a shallow neural network such that

$$\left\| \psi_k(\bar{\mathcal{P}}u) - \boldsymbol{A}_k\sigma\left(\boldsymbol{M}_k \cdot \bar{\mathcal{P}}u + \boldsymbol{b}_k\right) \right\|_{\infty} \leq Cm^{-1/d}, \tag{19}$$

where $\boldsymbol{a}_k^{\mathsf{T}} \in \mathbb{R}^m$, $\boldsymbol{M}_k \in \mathbb{R}^{m \times n}$, and $\boldsymbol{b}_k \in \mathbb{R}^m$. For the simplicity notations, we can replace $\boldsymbol{M}_k \cdot \bar{\mathcal{P}}$ by an operator $\mathcal{W}_k$.

Above all, we have that there is a neural network in $\Xi_p$ such that

$$\left\| \sum_{k=1}^{p} \boldsymbol{A}_k\sigma\left(\mathcal{W}_k u + \boldsymbol{b}_k\right)\sigma\left(\boldsymbol{w}_k \cdot \boldsymbol{x} + \zeta_k\right) - \mathcal{G}(u) \right\|_{L^2(\Omega)} \leq C_1 m^{-\frac{1}{n}} + C_2(\epsilon + p^{-\frac{2}{d}}), \tag{20}$$

where $C_1$ is a constant independent of $m$, $\epsilon$, and $p$, $C_2$ is a constant depended on $p$.

$\square$

---

[2] In this paper, we consistently employ the symbol $C$ as a constant, which may vary from line to line.

**Remark 3.** *We want to emphasize the reasonableness of our assumptions. For condition (i), we are essentially restricting our approximation efforts to local regions. This limitation is necessary because attempting to approximate the neural operator across the entire domain could lead to issues, particularly in cases where multiple solutions exist. Consider a scenario where the input function $u$ lies between two distinct solutions. Even a small perturbation of $u$ could result in the system converging to a completely different solution. Condition (i) ensures that Equation (2) has a unique solution, allowing us to focus our approximation efforts within localized domains.*

*Conditions (ii) and (iii) serve to regularize the problem and ensure its tractability. These conditions are indeed straightforward to fulfill, contributing to the feasibility of the overall approach.*

*For the embedding operator $\mathcal{P}$ in (iv), there are a lot of choices in DeepONet, such as finite element methods like Argyris elements [2] or embedding methods in [23, 18]. We will discuss more in the appendix. We omit the detailed discussion in the paper. Furthermore, for the differential neural network, this embedding may be different; for example, we can use Fourier expansion [43] or multigrid methods [16] to achieve this task.*

*Here, we discuss more about the embedding operator $\mathcal{P}$. One approach is to use the Argyris element [2]. This method involves considering the 21 degrees of freedom shown in Fig. 7. In this figure, each • denotes evaluation at a point, the inner circle represents an evaluation of the gradient at the center, and the outer circle denotes evaluation of the three-second derivatives at the center. The arrows indicate the evaluation of the normal derivatives at the three midpoints.*

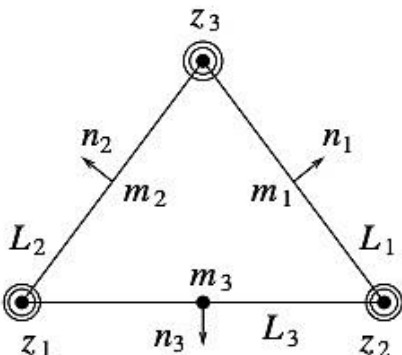

Figure 7: Argyris method

*Another alternative approach to discretizing the input space is to use the bi-cubic Hermite finite element method [23, 18].*

### B.3 Proof of Theorem 2

The proof of Theorem 2 is inspired by that in [22].

*Proof of Theorem 2.* **Step 1:** To begin with, we introduce a new term called the middle term, denoted as $\mathcal{E}_{Sm}(\boldsymbol{\theta})$, defined as follows:

$$\mathcal{E}_{Sm}(\boldsymbol{\theta}) := \frac{1}{M_u} \sum_{j=1}^{M_u} \int_{\Omega} |\mathcal{G}(u_j)(\boldsymbol{x}) - \mathcal{O}(u_j; \boldsymbol{\theta})(\boldsymbol{x})|^2 \, \mathrm{d}\boldsymbol{x},$$

This term represents the limit case of $\mathcal{E}_S(\boldsymbol{\theta})$ as the number of samples in the domain of the output space tends to infinity ($M_x \to \infty$).

Then the error can be divided into two parts:

$$|\mathbb{E}(\mathcal{E}_S(\boldsymbol{\theta}) - \mathcal{E}_{Sc}(\boldsymbol{\theta})| \leq |\mathbb{E}(\mathcal{E}_S(\boldsymbol{\theta}) - \mathcal{E}_{Sm}(\boldsymbol{\theta})| + |\mathbb{E}(\mathcal{E}_{Sm}(\boldsymbol{\theta}) - \mathcal{E}_{Sc}(\boldsymbol{\theta})|. \tag{21}$$

**Step 2:** For $|\mathbb{E}(\mathcal{E}_{Sm}(\boldsymbol{\theta}) - \mathcal{E}_{Sc}(\boldsymbol{\theta}))|$, this is the classical generalization error analysis, and the result can be obtained from [34, 42, 41]. We omit the details of this part, which can be expressed as

$$|\mathbb{E}(\mathcal{E}_{Sm}(\boldsymbol{\theta}) - \mathcal{E}_{Sc}(\boldsymbol{\theta}))| \leq \frac{Cd_{\boldsymbol{\theta}}\sqrt{\log M_x}}{\sqrt{M_x}}, \tag{22}$$

where $C$ is independent of the number of parameters $d_{\boldsymbol{\theta}}$ and the sample size $M_x$. In the following steps, we are going to estimate $|\mathbb{E}(\mathcal{E}_S(\boldsymbol{\theta}) - \mathcal{E}_{Sm}(\boldsymbol{\theta}))|$, which is the error that comes from the sampling of the input space of the operator.

**Step 3:** Denote

$$S_{\boldsymbol{\theta}}^M := \frac{1}{M}\sum_{j=1}^{M}\int_{\Omega}|\mathcal{G}(u_j)(\boldsymbol{x}) - \mathcal{O}(u_j;\boldsymbol{\theta})(\boldsymbol{x})|^2\,\mathrm{d}\boldsymbol{x}.$$

We first estimate the gap between $S_{\boldsymbol{\theta}}^M$ and $S_{\boldsymbol{\theta}'}^M$ for any bounded parameters $\boldsymbol{\theta}, \boldsymbol{\theta}'$. Due to Assumption 2 (i) and (ii), we have that

$$|S_{\boldsymbol{\theta}}^M - S_{\boldsymbol{\theta}'}^M|$$

$$\leq \frac{1}{M}\sum_{j=1}^{M}\left|\int_{\Omega}|\mathcal{G}(u_j)(\boldsymbol{x}) - \mathcal{O}(u_j;\boldsymbol{\theta})(\boldsymbol{x})|^2 - |\mathcal{G}(u_j)(\boldsymbol{x}) - \mathcal{O}(u_j;\boldsymbol{\theta}')(\boldsymbol{x})|^2\,\mathrm{d}\boldsymbol{x}\right|$$

$$\leq \frac{1}{M}\sum_{j=1}^{M}\left|\int_{\Omega}|2\mathcal{G}(u_j)(\boldsymbol{x}) + \mathcal{O}(u_j;\boldsymbol{\theta})(\boldsymbol{x}) + \mathcal{O}(u_j;\boldsymbol{\theta}')(\boldsymbol{x})| \cdot |\mathcal{O}(u_j;\boldsymbol{\theta})(\boldsymbol{x}) - \mathcal{O}(u_j;\boldsymbol{\theta}')(\boldsymbol{x})|\,\mathrm{d}\boldsymbol{x}\right|$$

$$\leq \frac{4}{M}\sum_{j=1}^{M}\Psi(u_j)\Phi(u_j) \cdot \|\boldsymbol{\theta} - \boldsymbol{\theta}'\|_{\ell^{\infty}}. \tag{23}$$

**Step 4:** Based on Step 3, we are going to estimate

$$\mathbb{E}\left[\sup_{\boldsymbol{\theta}\in[-B,B]^{d_{\boldsymbol{\theta}}}}\left|S_{\boldsymbol{\theta}}^M - \mathbb{E}S_{\boldsymbol{\theta}}^M\right|^p\right]^{\frac{1}{p}}$$

by covering the number of spaces.

Set $\{\boldsymbol{\theta}_1, \ldots, \boldsymbol{\theta}_K\}$ is a $\varepsilon$-covering of $[-B,B]^{d_{\boldsymbol{\theta}}}$ i.e. for any $\boldsymbol{\theta}$ in $[-B,B]^{d_{\boldsymbol{\theta}}}$, there exists $j$ with $\|\boldsymbol{\theta} - \boldsymbol{\theta}_j\|_{\ell_{\infty}} \leqslant \epsilon$. Then we have

$$\mathbb{E}\left[\sup_{\boldsymbol{\theta}\in[-B,B]^d}\left|S_{\boldsymbol{\theta}}^M - \mathbb{E}\left[S_{\boldsymbol{\theta}}^M\right]\right|^p\right]^{1/p}$$

$$\leq \mathbb{E}\left[\left(\sup_{\boldsymbol{\theta}\in[-B,B]^d}\left|S_{\boldsymbol{\theta}}^M - S_{\boldsymbol{\theta}_j}^M\right| + \left|S_{\boldsymbol{\theta}_j}^M - \mathbb{E}\left[S_{\boldsymbol{\theta}_j}^M\right]\right| + \left|\mathbb{E}\left[S_{\boldsymbol{\theta}_j}^M\right] - \mathbb{E}\left[S_{\boldsymbol{\theta}}^M\right]\right|\right)^p\right]^{1/p}$$

$$\leq \mathbb{E}\left[\left(\max_{j=1,\ldots,K}\left|S_{\boldsymbol{\theta}_j}^M - \mathbb{E}\left[S_{\boldsymbol{\theta}_j}^M\right]\right| + \frac{8\epsilon}{M}\left(\sum_{j=1}^{M}|\Psi(u_j)||\Phi(u_j)|\right)\right)^p\right]^{1/p}$$

$$\leq 8\epsilon\mathbb{E}\left[|\Psi(u_j)||\Phi(u_j)|^p\right]^{1/p} + \mathbb{E}\left[\max_{j=1,\ldots,K}\left|S_{\boldsymbol{\theta}_j}^M - \mathbb{E}\left[S_{\boldsymbol{\theta}_j}^M\right]\right|^p\right]^{1/p}. \tag{24}$$

For $8\epsilon\mathbb{E}\left[|\Psi\Phi|^p\right]^{1/p}$, it can be approximate by

$$8\epsilon\mathbb{E}\left[|\Psi\Phi|^p\right]^{1/p} \leqslant 8\epsilon\mathbb{E}\left[|\Psi|^{2p}\right]^{1/2p}\mathbb{E}\left[|\Phi|^{2p}\right]^{1/2p} = 8\epsilon\|\Psi\|_{L^{2p}}\|\Phi\|_{L^{2p}}.$$

For $\mathbb{E}\left[\max_{j=1,\ldots,K}\left|S_{\boldsymbol{\theta}_j}^M - \mathbb{E}\left[S_{\boldsymbol{\theta}_j}^M\right]\right|^p\right]^{1/p}$, by applied the result in [37, 22], we know

$$\mathbb{E}\left[\max_{j=1,\ldots,K}\left|S_{\boldsymbol{\theta}_j}^M - \mathbb{E}\left[S_{\boldsymbol{\theta}_j}^M\right]\right|^p\right]^{1/p} \leq \frac{16K^{1/p}\sqrt{p}\|\Psi\|_{L^{2p}}^2}{\sqrt{M}}.$$

**Step 5:** Now we estimate $|\mathbb{E}(\mathcal{E}_S(\boldsymbol{\theta}) - \mathcal{E}_{Sm}(\boldsymbol{\theta})|$.

Due to Assumption 2 and directly calculation, we have that

$$\|\Psi\|_{L^{2p}}, \|\Phi\|_{L^{2p}} \leqslant C(1 + \gamma\kappa p)^\kappa,$$

for constants $C, \gamma > 0$, depending only the measure $\mu$ and the constant $C$ appearing in the upper bound (7). For example,

$$\|\Psi\|_{L^{2p}} \leq \left( \int_{\mathcal{X}} C \left( 1 + \|u\|_{H^2(\Omega)} \right)^{2p\kappa} \mathrm{d}\mu_{\mathcal{X}} \right)^{\frac{1}{2p}}$$

$$\leq C \left( \int_{\mathcal{X}} \exp\left[ 2p\kappa \ln\left( 1 + \|u\|_{H^2(\Omega)} \right) - \alpha\|u\|_{H^2(\Omega)} \right] e^{\alpha\|u\|_{H^2(\Omega)}} \mathrm{d}\mu_{\mathcal{X}} \right)^{\frac{1}{2p}}$$

$$\leq C \left( \int_{\mathcal{X}} \left( 1 + \frac{\kappa p}{\alpha} \right)^{2\kappa p} e^{\alpha\|u\|_{H^2(\Omega)}} \mathrm{d}\mu_{\mathcal{X}} \right)^{\frac{1}{2p}} \leq C(1 + \gamma\kappa p)^\kappa. \tag{25}$$

Based on Lemma 1, we have that

$$\mathbb{E}\left[ \sup_{\boldsymbol{\theta}\in[-B,B]^{d_{\boldsymbol{\theta}}}} \left| S_{\boldsymbol{\theta}}^{M_u} - \mathbb{E}\left[ S_{\boldsymbol{\theta}}^{M_u} \right] \right|^p \right]^{1/p} \leqslant 16C^2(1 + \gamma\kappa p)^{2\kappa} \left( \epsilon + \left( \frac{CB}{\epsilon} \right)^{d_{\boldsymbol{\theta}}/p} \frac{\sqrt{p}}{\sqrt{M_u}} \right),$$

for some constants $C, \gamma > 0$, independent of $\kappa, \mu, B, d_{\boldsymbol{\theta}}, N, \epsilon > 0$ and $p \geqslant 2$. We now choose $\epsilon = \frac{1}{\sqrt{M_u}}$, so that

$$\epsilon + \left( \frac{CB}{\epsilon} \right)^{d_{\boldsymbol{\theta}}/p} \frac{\sqrt{p}}{\sqrt{M_u}} = \frac{1}{\sqrt{M_u}} \left( 1 + (CB\sqrt{M_u})^{d_{\boldsymbol{\theta}}/p}\sqrt{p} \right).$$

Next, let $p = d_{\boldsymbol{\theta}} \log(CB\sqrt{M_u})$. Then,

$$(CB\sqrt{M_u})^{d_{\boldsymbol{\theta}}/p}\sqrt{p} = \exp\left( \frac{\log(CB\sqrt{M_u})d_{\boldsymbol{\theta}}}{p} \right) \sqrt{p} = e\sqrt{d_{\boldsymbol{\theta}} \log(CB\sqrt{M_u})},$$

and thus we conclude that

$$\epsilon + \left( \frac{CB}{\epsilon} \right)^{d_{\boldsymbol{\theta}}/p} \frac{\sqrt{p}}{\sqrt{M_u}} \leqslant \frac{1}{\sqrt{M_u}} \left( 1 + e\sqrt{d_{\boldsymbol{\theta}} \log(CB\sqrt{M_u})}. \right).$$

On the other hand, we have

$$(1 + \gamma\kappa p)^{2\kappa} = \left( 1 + \gamma\kappa d_{\boldsymbol{\theta}} \log(CB\sqrt{M_u}) \right)^{2\kappa}.$$

Increasing the constant $C > 0$, if necessary, we can further estimate

$$\left( 1 + \gamma\kappa d_{\boldsymbol{\theta}} \log(CB\sqrt{M_u}) \right)^{2\kappa} \left( 1 + e\sqrt{d_{\boldsymbol{\theta}} \log(CB\sqrt{M_u})}. \right) \leqslant C \left( 1 + d_{\boldsymbol{\theta}} \log(CB\sqrt{M_u}) \right)^{2\kappa+1/2},$$

where $C > 0$ depends on $\kappa, \gamma, \mu$ and the constant appearing in (7), but is independent of $d_{\boldsymbol{\theta}}, B$ and $N$. We can express this dependence in the form $C = C(\mu, \Psi, \Phi) > 0$, as the constants $\kappa$ and $\gamma$ depend on the Gaussian tail of $\mu$ and the upper bound on $\Psi, \Phi$.

Therefore,

$$|\mathbb{E}(\mathcal{E}_S(\boldsymbol{\theta}) - \mathcal{E}_{Sm}(\boldsymbol{\theta})| \leq \mathbb{E} \sup_{\boldsymbol{\theta}\in[-B,B]^{d_{\boldsymbol{\theta}}}} |S_{\boldsymbol{\theta}}^{M_u} - \mathbb{E}\left[ S_{\boldsymbol{\theta}}^{M_u} \right]| \leq C\mathbb{E}\left[ \sup_{\boldsymbol{\theta}\in[-B,B]^{d_{\boldsymbol{\theta}}}} \left| S_{\boldsymbol{\theta}}^{M_u} - \mathbb{E}\left[ S_{\boldsymbol{\theta}}^{M_u} \right] \right|^p \right]^{1/p}$$

$$\leq \frac{C}{\sqrt{M_u}} \left( 1 + Cd_{\boldsymbol{\theta}} \log(CB\sqrt{M_u})^{2\kappa+1/2} \right). \tag{26}$$

$\square$

