# OpenReview forum: "Newton Informed Neural Operator for Solving Nonlinear Partial Differential Equations"
_NeurIPS.cc/2024/Conference — NeurIPS 2024 poster_

### Official Review · Reviewer_zTaB · 2024-07-10

**Soundness:** 2
**Presentation:** 1
**Contribution:** 3
**Rating:** 6
**Confidence:** 4

**Summary:**

The manuscript concerns an operator learning technique for elliptic partial differential equations with nonlinear, solution-dependent forcing terms. These PDE  can admit multiple solutions, which is a challenge for many existing PDE solution techniques, because they usually only result in a single solution. The authors propose to approximate the action of Newton's method, which is usually used to solve PDEs of this type, by a Deep-Operator-Network like architecture, so that the Newton steps can be performed more efficiently. They demonstrate the efficiency on multiple examples, and show results on convergence to the correct Newton iteration.

**Strengths:**

In general, the topic addressed in the manuscript is very interesting. Many machine learning techniques rely on the assumption that (at least...) there exists a unique solution to a problem (like a PDE here), and then approximate it as accurately as possible. The authors consider a much more challenging case: what if the problem has multiple solutions? They then proceed to use a classical numerical scheme (Newton's method) that is typically used to obtain one of the solutions of the given PDE, and approximate its action through a neural operator. This is interesting even away from the particular problem addressed here, because Newton's method is being used in many other applications as well.

**Weaknesses:**

Major:
1) The paper does not address the issue that "multiple solutions as output of the operator" effectively means "the input is mapped to a distribution of solutions". This framework has not been addressed adequately in the PDE learning community, but of course a very large portion of machine learning concerns exactly this problem: generative modeling (i.e., diffusion models, score based learning, image generation in general, etc., all try to "learn a distribution"). Effectively, a diffusion model is a "nonlinear operator" that maps an initial distribution to a final distribution (usually, in image space). The same is true even for simple variational autoencoders.

2) An even bigger issue is that there is no explanation as to why approximating the result of Newton's method leads to a "better" (in whatever way) approximation of PDEs with *multiple* solutions. Newton's method is deterministic, and there is no guarantee that it will eventually (for some reason) "sample" all solutions. The introduced "Newton Operator Network" architecture seems to approximate the action of Newton's method on a given state, so that the operation can be performed more efficiently after the operator is trained. It does not seem to help to find multiple solutions.

3) The approximation of Newton's method has been addressed in earlier work already, which is not cited.

 [A] Doncevic, D.T., Mitsos, A., Guo, Y., Li, Q., Dietrich, F., Dahmen, M., Kevrekidis, I.G., 2024. A Recursively Recurrent Neural Network (R2N2) Architecture for Learning Iterative Algorithms. SIAM J. Sci. Comput. 46, A719–A743. https://doi.org/10.1137/22M1535310

 [B] Chevalier, S., Stiasny, J., Chatzivasileiadis, S., 2022. Accelerating Dynamical System Simulations with Contracting and Physics-Projected Neural-Newton Solvers, in: Proceedings of The 4th Annual Learning for Dynamics and Control Conference. Presented at the Learning for Dynamics and Control Conference, PMLR, pp. 803–816.



Minor:
1) l.30: it is also important to mention that operator learning techniques typically require a training data set, while methods that directly solve the PDE typically do not.
2) l.37: "Newton methods provide well-defined locally" is missing a word
3) l.50: "solve the inverse problem" is not clear at this point. Which inverse problem?
4) The writing could be improved in general, with some expressions like "As mentioned earlier" (l.41) and "The following paper is organized as follows" (l51) being unnecessarily complicated. Section 2 "Backgrounds and Relative Works" should probably be "Background and related work". I do not list all language mistakes here.
5) Section 2.1 is not necessary, the information can be incorporated more concisely in the introduction.
6) l106,107: the index "i" is not used in the equation, same with "u_0" and "u dash".
7) l140 P_dash is not explained.
8) l172: this is not a complete sentence.
9) l244: L(u) cannot be $-\Delta u-u^2$, because it has to be a linear operator (as defined in line 20 of the manuscript). The example is still admissible, because the term $u^2$ can be absorbed into f(u).
10) l246-l248 contain multiple mistakes in the sentence structure.

**Questions:**

1) How does the approximation of the action of Newton iterations lead to a sampling of "all" (or at least more than one) solution of the PDE?

2) What are other methods that can be used to obtain multiple solutions to the given class of PDE, and what would it entail to learn them? Are they slower than the solution proposed here?

3) In case a specific PDE has to be solved (ideally, with several of its solutions as a result), does it make sense to train the Newton operator first, or would it be more reasonable to just use the classical Newton's method, because it can be started immediately without training the network?

**Limitations:**

The main paper does not contain any statements on specific limitations, or a broader discussion on potential negative impact of the work. This must be addressed.

---

> ### Author Rebuttal · Authors · 2024-08-05
>
> ## Weaknesses
>
> > Approximation of Newton's Method and Finding Multiple Solutions
>
> - Thank you for your insightful comments.
> We acknowledge that a large portion of machine learning research indeed focuses on learning distributions, as seen in generative models like diffusion models, score-based learning, and variational autoencoders. These models typically aim to map an initial distribution to a final distribution, often in a high-dimensional space like images.
>
> - However, we would like to clarify that our method, the Newton Informed Neural Operator, differs from these generative modeling approaches. Our approach is not designed to learn or output a distribution of solutions per se. Instead, it focuses on efficiently approximating the deterministic action of Newton's method on a given initial state, specifically in the context of solving nonlinear PDEs.
>
> - Our primary goal is to enhance the computational efficiency of finding multiple solutions by allowing for extensive exploration of initial conditions. The primary advantage of our approach lies in its efficiency; This efficiency gain is due to the neural operator's ability to quickly approximate the solution to each linear system, leveraging the pre-trained network to perform these tasks much faster than traditional solvers. The details of how we benchmarked the efficiency are provided in the paragraph **Efficiency** and in Appendix A.5 **Benchmarking Newton’s Method and Neural Operator-Based Method**. This significant speed-up allows for the practical exploration of a much larger space of initial conditions in a reasonable timeframe. By efficiently sampling a wide variety of initial states, our method increases the likelihood of finding multiple solutions of a PDE, especially in cases where the solution landscape is complex.
>
> >  Prior Work on Newton's Method Approximation
>
> We appreciate the reviewer's insight into prior work that also addresses the approximation of Newton's method using neural networks. We will ensure to include the following references in the revised manuscript:
>
> - While these cited works focus on ODEs and propose neural network architectures like R2N2 to emulate iterative methods such as Runge-Kutta and Krylov, our paper extends these concepts to handle partial differential equations (PDEs). This extension involves significant additional considerations including spatial discretization, boundary conditions, and more complex solution structures.
>
> - Our contribution specifically introduces the **Newton Informed Neural Operator**, which incorporates a **Newton informed loss function**. This function leverages classical Newton methods to refine the approximation capabilities of neural networks for PDEs. This approach is distinct in that it is tailored to capture multiple solutions and managing spatial complexity.
>
> - We believe that while the foundational ideas may overlap, our work provides a novel application and significant advancements in the context of PDEs. We will clearly delineate these differences in our revision and appropriately contextualize our contributions in relation to the existing literature.
>
> > Typos and Expressions
>
>  Thank you for your careful reading. We will correct the typos and inappropriate expressions in the manuscript. Specifically:
>   - In line 172, the term $P_{dash}$ will be replaced with a rigorous expression. Please also refer to our response to Reviewer 6H5E for additional details.
>   - In line 244, $L(u)$ should be corrected to $-\Delta u$.
>
> ## Questions
>
> We believe all three questions are discussing the rationale for training a neural operator to approximate Newton's method. We list it in the following:
>
> - There exists no general method with a guarantee that all the solutions for complex nonlinear problems can be found, for example, the Gray Scott model. Newton's method is a very efficient method but suffers from the computational cost, especially when extensively sampling from various initial conditions as we discussed in the weaknesses section.
>
> - Once the neural operator is trained, this efficiency gain is due to the neural operator's ability to quickly approximate the solution to each linear system, leveraging the pre-trained network to perform these tasks much faster than traditional solvers. The details of how we benchmarked the efficiency are provided in the paragraph **Efficiency** and in Appendix A.5 **Benchmarking Newton’s Method and Neural Operator-Based Method**.
>
> - For a simple problem such as the convex problem in our **Case 1**, the extra time for data generation and training makes the neural operator-based method less useful, but for a complex case like the Gray Scott model, the efficiency gain of a pre-trained neural operator compensates for the cost of training.
>
> ## Limitations and Broader Impact:
> We admit that limitations are not discussed enough. The limitations include two main parts:
>   - We did not conduct baseline comparisons or ablation studies, as our primary intent was to prove the concept that neural operators can be effectively used to solve PDEs with multiple solutions. Future work could explore these aspects in greater detail, comparing different neural operator architectures and their performance relative to traditional numerical methods.
>   - Although the Newton informed loss function alleviates the cost for data generation, the data efficiency of the method still needs to be improved.

---

> > ### Comment · Reviewer_zTaB · 2024-08-10
> >
> > I thank the authors for the careful revision. The explanation in the general comment was very helpful as well - it seems the major issue was the presentation of the results, not the idea or results themselves. I read the paper again and I am now more excited about the idea and results. Indeed, learning Newton's method in this context makes a lot of sense - if the classical approach requires millions of calls to a Newton solver (for the millions of initial conditions), then replacing this inner loop with a learned model is great. I updated my score to 6.

---

> > > ### Author Response · Authors · 2024-08-10
> > >
> > > We are pleased that the revisions have clarified the novelty of our approach. We appreciate your recognition of the potential of our method. Thank you once again for your valuable insights and constructive discussion.

---

### Official Review · Reviewer_3WGd · 2024-07-11

**Soundness:** 3
**Presentation:** 2
**Contribution:** 4
**Rating:** 5
**Confidence:** 4

**Summary:**

The authors presented a new machine learning based technique called the Newton-informed Neural Operator for solving PDEs that have multiple solutions due to nonlinearities. This is an important problem, and the Newton informed neural operator is capable of obtaining these multiple solutions using a single, unified training process. The authors present experimental evidence showing that their method indeed is capable of recovering these multiple solutions.

**Strengths:**

The fundamental idea behind this work is timely, given the rise of ML-based techniques to solve PDEs, including but not limited to physics-informed neural nets (PINNs) and operator learning. ML is a promising way of obtaining multiple solutions to PDEs, and the Newton informed operator appears to do this quite well.

**Weaknesses:**

Unfortunately, this paper lacks clarity of exposition and the Newton-informed Neural operator is never clearly written down nor is its architecture illustrated anywhere. It appears to be a standard DeepONet augmented with a specific loss function, but the derivation of that loss function is unclear. I also don't see clearly how Equation 2 represents Newton's method, since a proper linearization would produce Jacobian terms involving both L and f. A derivation would have been helpful, but this is missing. The authors also do not compare against HomPINNs, which are designed for learning multiple solutions; if these can be used only in a supervised setting, why not compare in that setting? Finally, in addition to these serious weaknesses, I also could imagine other ways of solving this problem (such as an interval Newton iteration on a standard DeepONet).

**Questions:**

1. Introduction on page 1: are the multiple solutions coming from different possible choices of the coefficient functions, or just from the nonlinearity?  The introduction says "singular" solution when they mean unique solutions. This is somewhat problematic, since "singular" also has a mathematical meaning. I would reword.
2. Line 37 page 2, "Newton methods provide well-defined locally": is there a missing word?
3. Line 42 page 2, "Our approach combines the classical Newton method, which...": combines it with what?
4. Lines 92-95, page 3: I'm struggling to understand what the authors are saying by "remains uncontrollable". Reword for clarity?
5. Lines 99-103, page 3: Is this true? If a solution exists, there must be an operator, surely? Citations and/or proof needed.
6. Equation 2: How is this Newton's method? I see no linearization of the nonlinear operator, no Jacobian term corresponding to the nonlinearity, and only an f'(u). Perhaps I'm missing something.
7. Page 4, Line 126: "In this paper, we will use DeepONet to approximate the Newton operator". What exactly is the Newton operator? None of the operators written out so far have been called that.
8. Line 130, page 4: "Furthermore, MgNO is replaced by...". This whole sentence is odd. Are the authors trying to say that different choices of W lead to different operator learning methods? If so, this makes sense but needs rewording.
9. Page 4, remark 1: DeepONet hasn't even been described yet. How are general readers supposed to know how this fits into this paper?
10. Page 5, Section 3.3: I still haven't seen an architecture or description of the Newton-informed Neural operator, but we're already talking about a loss function.
11. Page 6, Section 3.3.2: We finally get to the "Newton loss". It looks like a Newton-informed operator is a DeepONet trained with the Newton loss? But where did the loss in Equation 9 come from, how was it derived? I don't see a clear connection to the previous exposition.

**Limitations:**

The authors do not explicitly have a limitations section, but they briefly address their limitations in their conclusion.

---

> ### Author Rebuttal · Authors · 2024-08-05
>
> ## Weaknesses
>
> > Clarity of Exposition:
> - We will include a clear architectural diagram of the Newton-informed Neural Operator and provide a step-by-step derivation of the Newton loss function as follows:
>
> For the nonlinear PDEs:
> $$
> \begin{cases}
> \mathcal{L} u(\mathbf{x}) = f(u), & \mathbf{x} \in \Omega \\
> u(\mathbf{x}) = 0, & \mathbf{x} \in \partial \Omega,
> \end{cases}
> $$
> we can represent the PDEs as $ F(u) = 0 $, where $ F $ is an operator defined as $ F(u) = \mathcal{L}(u) - f(u) $. In the context of the Newton method for operators, we start with an initial condition $ u_0 $ and iteratively obtain a solution. For each iteration, $ u_{n+1} = u_n + \delta u_n $, where $ u_n $ is the solution from the previous iteration, and $ \delta u_n $ is obtained from the iteration equation $ F'(u) \delta u = F(u) $. Here, $ F'(u) $ is the (Fréchet) derivative of the operator, defined as follows:
>
> To find $ F'(u) $ such that for any $ v \in H^2(\Omega) $,
> $$
> \lim_{\|v\| \to 0} \frac{\|F(u + v) - F(u) - F'(u)v\|}{\|v\|} = 0,
> $$
> where $ \|\cdot\| $ is the norm in the Sobolev space $ H^2(\Omega) $. In our case,
> $$
> F'(u)v + \mathcal{O}(v) := \mathcal{L}(u + v) - \mathcal{L}(u) - (f(u + v) - f(u)) = \mathcal{L}(v) - f'(u)v + \mathcal{O}(v).
> $$
>
> Therefore, the Newton linear PDE we use in this paper is given by:
> $$
> \begin{cases}
> (\mathcal{L} - f'(u)) \delta u(\mathbf{x}) = \mathcal{L} u - f(u), & \mathbf{x} \in \Omega \\
> \delta u(\mathbf{x}) = 0, & \mathbf{x} \in \partial \Omega.
> \end{cases}
> $$
> This represents the linear PDE for $ \delta u $. Based on the assumptions in our paper, we can prove that this has a unique solution.
>
> In this paper, we aim to solve nonlinear PDEs with any given initial guess. Therefore, we need to learn the Newton solver, which is an operator between $ u $ and $ \delta u $. The way to learn this is by employing DeepONet with a Newton information loss. The structure of DeepONet is provided in the appendix, and the Newton information loss is derived from the above equation. The Newton loss function is defined as:
> $
> E_{N}(\theta) := \frac{1}{N_u \cdot N_x} \sum_{j=1}^{N_u} \sum_{k=1}^{N_x} \left|(\mathcal{L} - f'(u_j(x_k))) \mathcal{O}(u_j; \theta)( x_k) - \mathcal{L} u_j(x_k) - f(u_j(x_k))\right|^2
> $
> where $ u_1, u_2, \ldots, u_{N_u} \sim \nu $ are independently and identically distributed (i.i.d.) samples in $ \mathcal{X} $, and $ x_1, x_2, \ldots, x_{N_x} $ are uniformly i.i.d. samples in $ \Omega $. As you can see, this loss function directly follows from the Newton iteration.
>
> > Comparison with HomPINNs:
> HomPINNs are developed to compute multiple solutions of nonlinear PDEs. In this paper, we present an operator learning approach combined with Newton's method to learn the nonlinear solver. While this approach is not specifically designed for computing multiple solutions, it can accelerate the nonlinear solver process if initial guesses are provided.
>
> ## Questions
>
> > Multiple Solutions and Nonlinearity:
> - In this paper, we focus on the multiple solutions arising from nonlinear terms in nonlinear PDEs. Analyzing the energy formula reveals that such terms may contain multiple local minima, leading to multiple solutions, commonly referred to as pattern formation. In contrast, for linear PDEs with multiple solutions, the differences are typically in the form of a constant $C$ or other simple polynomial functions. These simpler cases are not the focus of our study.
>
> > Methodological Clarifications:
> - Thank you for the careful reading. We will correct the typos in the paper and the details of our method as shown above. Additionally, we will move the structure of DeepONet from the appendix into the introduction.

---

> > ### Comment · Reviewer_3WGd · 2024-08-12
> >
> > This makes a lot of things much clearer. Assuming that the authors commit to revising the paper with the details above, I raise my score.

---

> ### Author Response · Authors · 2024-08-10
>
> We would like to further clarify the novelty of our method:
>
> 1. We introduced a general computational framework (as illustrated in the global response) for learning the Newton's nonlinear solver using neural operators integrated with the Newton-informed loss, with DeepONet/DeepPOD (architecture detailed in Appendix A.3) as a representative example. This computational framework can be used to identify multiple solutions of nonlinear PDEs.
>
> 2. After a thorough review of the HomPINNs paper, we identified a key distinction between our method and theirs. In HomPINNs, networks are primarily used to approximate/parametrize the solution. In contrast, our method leverages networks to approximate the Newton solver—specifically, the mapping from $u_k$ to $\delta u$,  $u_{k+1} = u_k+ \delta u$. While HomPINNs are designed to compute multiple solutions starting from simple initial functions, typically on coarse grids, by combining the homotopy approach with PINNs to learn multiple solutions on finer grids, our Newton-informed operator learning method directly trains the nonlinear solver on fine grids. This approach offers greater efficiency and reduces the risk of overlooking solutions that might be missed by HomPINNs.

---

### Official Review · Reviewer_tGuq · 2024-07-12

**Soundness:** 2
**Presentation:** 2
**Contribution:** 2
**Rating:** 5
**Confidence:** 2

**Summary:**

In this paper, the authors propose a newton informed neural operator for solving PDEs. Based on the description, it seems the neural operator is designed to predict the solution to the PDE given an initial solution (equation below eq. 2 on pg 3). The neural operator is trained with 2 types of losses, mse loss and newton loss. In the newton loss, the linearized PDE as shown in Eq. 2 is minimized using the neural operator predictions. The method is demonstrated on several use cases.

**Strengths:**

The method claims to learn multiple PDE solutions given any initial solution and uses a newton based loss for improved learning.

**Weaknesses:**

The paper is hard to follow. The training and inference workflows of the DeepONet are unclear. The generation of the data is not clearly described. There are no baseline comparisons or ablation studies to understand and analyze the method in a better way.

**Questions:**

- The authors mention 500 supervised samples are generated but corresponding to what conditions? How is the testing data different from the training data? What happens in the scenario where the initial solution is slightly perturbed, how does the prediction and converged solution from the newtons solver compare?
- Error metrics between the solution generated by the newtons solver and the operator predictions are need to be provided for the solution generated. How do you validate the correctness of the solutions generated from the neural operator?
- In the results reported in Table 1, how does the neural operator take only 1e-4 seconds to perform 500 or 5000 iterations? Is it because it is converging faster? The information presented in the table is unclear.
- Based on the time reported in Table 1 indicate that the neural operator can directly predict the solution of the PDE given any initial condition whereas the newton solver has to iteratively solve it. Or, is the neural operator also recursively predicting the solutions? If so is this recursive operation performed while computing the Newtons loss as well? How is the memory managed in that case? Many of these details are unclear in the paper.
- The newton loss requires the computation of the adjoint and solution vector product which can be highly memory and compute intensive as the size of the grid increases? Can the authors provide details on how their method scales with increasing grid sizes?
- What is deepPOD network in Figure 2? Does it refer to the POD calculation in the trunk net? How will this POD calculation be carried out for a grid with million cells? Would it blow up the size of the trunk net? Would this make the neural operator evaluation slower as the grid size increases?
- How do results from method 1 and method 2 compare for the different use cases?

---

> ### Author Rebuttal · Authors · 2024-08-05
>
> ## Weaknesses
>
> > Data Generation:
> - For method 1, we use 500 supervised data samples (with ground truth), while for method 2, we use 5000 unsupervised data samples (only with the initial state) along with supervised data samples.
>
> Here is how we generate the supervised data samples:
>
> 1. **Step 1**: We use a classical numerical solver to obtain single (multiple) solutions of nonlinear PDEs. For example, in **Case 2**, there exist four solutions $u^1, u^2, u^3, u^4$. We have one solution for **Case 1** and 10 solutions for the **Gray-Scott model**.
>
> 2. **Step 2**: The supervised dataset is generated by sampling a perturbation around the solution $u^i_{p} \sim \mathcal{N}(0, (-\Delta)^{-3})$ on the chosen true solution $u^i$. We then set $u^i_0 = u^i_{p} + u^i$ and calculate the convergent sequence $\{u_0^i, u_1^i, \ldots, u_n^i\}$ using Newton's method, which follows the formula:
>    $$
>    u_{k+1}^i = u_k^i - J_f(u_k^i)^{-1} f(u_k^i).
>    $$
>    Each convergent sequence $\{u_0^i, u_1^i, \ldots, u_n^i\}$ constitutes one supervised data entry in the dataset.
> In this case, we consider the initial conditions as perturbed, with all perturbations applied around the true solution. A comparison between the traditional method and our proposed method is summarized in Table 1.
>
> > Baseline Comparisons and Ablation Studies:
> - We acknowledge the reviewer's concern regarding the lack of baseline comparisons and ablation studies. However, the primary goal of our work is to establish that neural operators can effectively solve partial differential equations (PDEs) with multiple solutions. Our aim is to demonstrate the feasibility and potential of this approach rather than to provide an exhaustive comparison with existing neural operators.
>
> - To this end, we provided a general framework for finding multiple solutions using neural operators and used the DeepPOD as an example to showcase the applicability of our method to several PDE problems with multiple solutions. Our focus was on illustrating that various neural operators, such as the Fourier Neural Operator, can be seamlessly integrated into our framework, replacing the DeepOnet/DeepPOD. This demonstrates the versatility and robustness of our approach.
>
>
>
> ## Questions
>
> >Data Generation and Testing Conditions:
> - We have addressed the procedures in the weakness regarding how to generate 500 supervised samples. As for the test data and training data, their initial states are both sampled from the same distribution $u^i_{p} \sim \mathcal{N}(0, (-\Delta)^{-3})$.
>
> - If the initial state is only slightly perturbed, the Newton informed neural operator will converge quickly as the classical Newton's method, since the Newton informed neural operator approximates each Newton's step and recursively predicts the true solution. We compare the error for each step with Newton's method in Figure 1 and the error for the final prediction in Figure 3.
>
> > Error Metrics and Validation:
> - The error is measured by $L^2$ and $H^1$ norm, for example in Figure 1 and Figure 3. The definitions are provided in Appendix B.1.
>
> >The Information Presented in the Table is Unclear:
> - The time reported in Table 1 reflects the efficiency of the neural operator compared with classical direct solvers for Newton's linear systems. It is not because the neural operator converges faster in the sense of fewer iterations. Rather, the time is measured for solving 500 or 5000 independent Newton's linear systems, each with different initial states.
> - This efficiency gain is due to the neural operator's ability to quickly approximate the solution to each linear system, leveraging the pre-trained network to perform these tasks much faster than traditional solvers. The details of how we benchmarked the efficiency are provided in the paragraph **Efficiency** and in Appendix A.5 **Benchmarking Newton’s Method and Neural Operator-Based Method**. These sections explain the experimental setup and the metrics used to evaluate the performance.
>
> > Does the Neural Operator Recursively Predict the Solution? How does the Method Scale with Grid Size.
> - Neural operator recursively predict the solution, since the Newton informed neural operator approximates the Newton's iteration step at given state. Since the architecture is simple, the memory is only of  $\mathcal{O}(N)$  in terms of the degrees of freedom $N$.
> - At the evaluation phase of the Newton informed neural operator-based solver, the computational cost is $\mathcal{O}(N)$ while the classical direct solver for solving the Newton's step is $\mathcal{O}(N^3)$ for dense systems and less for sparse systems, but still significant.
> - The POD calculation does indeed scale with the grid size, and for grids with millions of cells, this could lead to increased computational costs. In such cases, alternative approaches, such as using a pre-defined mesh-less basis like kernel basis functions, can be utilized to manage computational complexity and maintain efficiency.
>
> >  DeepPOD Network Architecture?
> - Yes, the deepPOD network in Figure 2 refers to the neural operator where the trunk net is replaced by pre-calculated basis functions obtained via the Proper Orthogonal Decomposition (POD) method. The discussion regarding the architecture is discussed in Appendix A.3.
>
> We want to emphasize that our framework is not tied to a specific neural operator architecture. While we demonstrated our method using the deepPOD as an example, our primary claim is that neural operators, in general, can effectively find multiple solutions for partial differential equations. For different PDEs or larger grid sizes, more advanced neural operator architectures could be employed to achieve optimal performance.

---

> ### Author Response · Authors · 2024-08-12
>
> Dear Reviewer,
>
> As the discussion period draws to a close, we kindly invite you to reassess your review in light of our responses. Please let us know if you have any remaining questions or concerns.
>
> Best wishes,
> Authors

---

> > ### Comment · Reviewer_tGuq · 2024-08-13
> > **Comments on rebuttal**
> >
> > I would like to thank the authors for their revision. I understand that training the neural operator with a Newton's loss is novel and the main goal of the paper is to demonstrate that neural operators can be trained for multiple solutions but I have a few followup questions.
> >
> > If the training and testing data set is generated by considering small perturbations on 10 solutions then, is the validity of the trained model only within these bounds of the initial solutions? The authors claim that " This significant speed-up allows for the practical exploration of a much larger space of initial conditions in a reasonable timeframe." Where is this claim verified in the paper? Is the testing carried out on initial conditions + perturbations different from the 10 considered in the Gary Scott use case?
> >
> > The errors reported in Figs. 1 and 3 are about 1-2 orders of magnitude larger than the training errors. Is this considered as a reasonable error? Why is that the case when the testing and training samples are from the same distribution? Do the testing results improve if you increase the number of training samples?
> >
> > The neural operator evaluation is recursive meaning a solution at "n+1" is calculated based on the solution at "n". Table 3 reports that 500 evaluation steps require 1.1e-4 seconds and for 5000 evaluation steps the time increases to 1.4e-4. Why is that the case? Why doesn't the time increase 10x like the Newtons solver? The time taken by each step is constant and proportional to the number of parameters in the NN and by that logic it should be more for 5000 steps vs 500 steps. Or are these operations batched meaning you are predicting all the 500 solutions at once? If that is the case then it is not a correct representation because the solutions at future iterations are unknown during inference. Am I missing something?

---

> > > ### Author Response · Authors · 2024-08-13
> > >
> > > We sincerely thank the reviewer for the insightful feedback.
> > >
> > > 1. The supervised training data includes perturbations on 10 solutions, representing only a small portion of the entire training dataset. The unsupervised data, on the other hand, is sampled from a much larger space with greater variance. This intentional setup is to demonstrate that only supervised data is not enough (also related with your second question "why test error is much larger than training error").  Our model is not limited to small perturbations around known solutions but is also effective in larger space. This effectiveness is due to the unsupervised data being sampled and trained using Newton's loss function economically. This is the primary motivation for using Newton's loss. In lines 287-290 and Figure 3(a), we show that our model, trained with unsupervised data, can successfully handle pattern formations that are not present in the supervised dataset. In that figure, the test case (a ring pattern) differs from the 10 patterns considered in the supervised dataset, but rather close to unsupervised training dataset.
> > >
> > > 2. Your questions align precisely with what we aim to demonstrate in Figure 1 and lines 237-241. In Figure 1(a), the significantly larger testing error compared to the training error suggests that using only MSE loss is insufficient (model is only valid at small pertubation around 10 solutions in this case). Figure 1(b) illustrates that increasing the number of unsupervised samples leads to a substantial improvement in the test L2 error, highlighting the necessity of introducing Newton's loss. The testing samples are drawn from the same distribution as the unsupervised training samples.
> > >
> > > 3. These operations are parallelized, meaning that we predict all 500 solutions simultaneously. The parallelization of solving 500/5000 Newton steps, whether using a classical Newton solver or a neural operator, is reasonable because we are addressing 500/5000 problems with 500/5000 different initial conditions. For a fair comparison, the classical Newton solver is also parallelized using CUDA implementation on a GPU. The speedup of the neural operator arises from its ability to benefit more from batch processing, as it naturally handles large batch sizes efficiently during the inference phase. We discuss this in more detail in Appendix A.5. This leads back to our primary motivation: enhancing the computational efficiency of finding multiple solutions by enabling extensive exploration of initial conditions. This significant speedup allows for practical exploration of a much larger space of initial conditions within a reasonable timeframe. We further discuss our motivation and methodology in the global response and response to Reviewer zTaB.
> > >
> > > We thank the reviewer again for the valuable discussion and will revise the paper accordingly to improve the clarity.

---

> > > > ### Comment · Reviewer_tGuq · 2024-08-13
> > > > **Official comment**
> > > >
> > > > Thanks for the clarification.
> > > >
> > > > For example, if we look at Figure 3, the training loss is around 0.01 while the testing is of the order 1.0. I completely understand that the testing error improves with the addition of the Newton loss but my main question is whether that improvement is sufficient considering this discrepancy between the train and test loss? If the testing errors are large then how can this method be used to explore a range of initial conditions as the authors claim? I understand that the ring pattern shown in Fig. 3 is completely unsupervised. Are there other unsupervised initial conditions that have been analyzed? What guarantees that the method will always converge and the testing errors will remain bounded for any unsupervised initial condition?
> > > >
> > > > I think the text in Table 1 needs to be modified because it is confusing. It gave me the impression that you are solving 1 initial condition but performing 500 and 5000 linear iterations in the Newton's solver. However, it seems like you are solving 500 or 5000 initial conditions until convergence.

---

> > > > > ### Author Response · Authors · 2024-08-13
> > > > >
> > > > > Thank you for your comments.
> > > > >
> > > > > In Figure 3, the training loss and the absolute L2 test error (which is large because the magnitude of the true solution is large) are not directly comparable, as they represent different metrics. The training loss reflects the rescaled Newton's loss, as we have demonstrated. Therefore, their magnitudes should not be compared directly. The reason we plot them together is to illustrate that their trends during the training process are consistent. This consistency suggests that the unsupervised data, when trained with Newton's loss, contributes to the improvement of the model's performance. We will revise the figure to avoid any confusion.
> > > > >
> > > > > Regarding convergence, it is important to note that we cannot guarantee it, as even the exact Newton's method does not ensure convergence for every unsupervised initial condition. For other unsupervised initial conditions, if the unsupervised data contains the solution and has enough data points, we can achieve convergence by effectively training with the Newton loss. However, if there is not enough unsupervised data for certain types of solutions, we cannot guarantee the same level of success. Nonetheless, our experiments demonstrate that adding unsupervised data expands the exploration range of initial conditions.
> > > > >
> > > > > We will also enhance the clarity of Table 1 in the revised version.

---

> > > > > > ### Comment · Reviewer_tGuq · 2024-08-13
> > > > > > **Official comment**
> > > > > >
> > > > > > Thanks for the clarification. Please make the necessary revisions to the paper to improve the presentation of the ideas and results. I have increased my score to 5.

---

> > > > > > > ### Author Response · Authors · 2024-08-13
> > > > > > >
> > > > > > > We appreciate your acknowledgment of the potential of our method. We will revise our paper based on our discussion. Thank you once again for your valuable insights and constructive feedback.

---

### Official Review · Reviewer_6H5E · 2024-07-14

**Soundness:** 3
**Presentation:** 3
**Contribution:** 3
**Rating:** 5
**Confidence:** 3

**Summary:**

The paper proposes the Newton Informed Neural Operator, a novel method that leverages classical Newton methods and neural network techniques, to efficiently learn multiple solutions to nonlinear PDEs.

Classical Newton's method iteratively linearizes the nonlinear equation and solves the resulting linear system to find one solution. The proposed Newton Informed Neural Operator incorporates DeepOnet to learn the operator solution to these linearized equations. After training, the nth iteration of the trained neural operator will approximate one solution.

The paper demonstrates the effectiveness of the proposed Newton Informed Neural Operator through experiments on various nonlinear PDEs. Notably, it accurately captures multiple solutions with a significant reduction in computational cost compared to traditional methods.

**Strengths:**

The paper introduces a novel methodology that combines classical Newton methods with neural network techniques to address the challenge of computing multiple solutions of nonlinear PDEs.

The paper provides complete theoretic results, including approximation and generalization error analysis.

The methodology presented in the paper has wide applicability in fields such as physics, biology, and engineering, where nonlinear PDEs with multiple solutions are common. By efficiently handling such problems, this approach can contribute to advancements in various scientific disciplines.

**Weaknesses:**

1. The paper may lack a comprehensive comparison with existing methods. It is stated that applying the classical solver to solve Eq. (1) multiple times can be time-consuming, however,  the proposed method requires date generation, NN training, and several iterations of the trained neural operator. Without a thorough comparison, it may be challenging for readers to gauge the novelty and effectiveness of the proposed method.

2. The symbols used in the text are somewhat confusing and require careful revision.
2.1. In Section 3.1, $\tilde{u}=u+\delta u$ is obtained by solving Equation 2, however, $u^*$ is the solution of Equation 2 in Section 3.2.
2.2. In the fourth term of Assumption 1, how do we define the H^2 norm of u-Pu? What is $\bar{P}u$? The reviewer guesses that the paper may be describing something similar to the Schauder basis and canonical projections. But the wording here is not rigorous.
2.3 What is the definition of $f(u)$ and $f'(u)$. Is $f$ just a function like $u^2$ rather than an operator?

3. The method  only needs to learn the operator solution to the linearized equations. It seems that this is a task that can be completed by a standard DeepOnet. Is there any difference between the neural network part and the approximation and generalization analysis used in this paper and those of the existing DeepOnet?

**Questions:**

1. The paper highlights that the proposed method requires fewer computational times when solving multiple linear Newton systems (e.g., 500 or 5,000 systems). This is a significant advantage. What if it's just a single linear Newton system? Do we really need to solve multiple linear Newton systems simultaneously here? In this paper, the goal is to solve linear Newton systems sequentially, correct? Could you also explain the aim of Table 1?

2. Can the paper demonstrate the computational time required to solve the 4.4 Gray-Scott model using classical Newton iteration and the method proposed in this paper? as well as their accuracy?

3. Section B.2: Is there any specific difference between the approximation ability of the DeepOnet proved in the text and the approximation ability of the general DeepOnet. It seems that the key steps 3, 4 and 5 do not involve specific operators.

4. It is stated that equation 18 is derived via proposition 1 in subsection 16 in [26] (H. Mhaskar, Neural networks for optimal approximation of smooth and analytic functions. Neural computation, 8(1):164–177, 1996).  Is this a typo? It seems that the reference mentioned does not have such a proposition.

5. Does the considered embedding operator using Argyris element satisfy Assumption 1.4? Could the paper provide a rigorous mathematical description?

6. Theorem 2: Does the symbol E refer to expectation? Could you write it out in full in integral form? Does the theorem 2 hold for any parameter $\theta$?

7. How does the paper obtain equation 25? Is there a typo?

**Limitations:**

Yes

---

> ### Author Rebuttal · Authors · 2024-08-05
>
> ## Weaknesses
>
> > Comparison:
>
> - The idea behind our proposed Newton Informed Operator Learning is rooted in the computational data generated by classical Newton's methods. When computing multiple solutions of nonlinear PDEs in pattern formation, the initial guesses can be in the millions, especially with fine grid solutions. In such cases, we utilize a subset of the data generated by the classical Newton's method to train the Newton Informed Operator. This training process is performed offline. Once the operator is well-trained, it can be applied to replace the classical solver, significantly speeding up the computation.
>
> - We agree that a more comprehensive comparison with existing methods could further strengthen our paper. However, our primary goal was to highlight the novel integration of classical Newton methods with neural network techniques for solving nonlinear PDEs. We have demonstrated the speedup by comparing it with traditional Newton methods. To the best of our knowledge, no existing methods specifically address the acceleration of nonlinear PDE computation in the way we propose.
>
> > Clarity of Notation:
>
> - 2.1 The use of symbols will be clarified in the revised manuscript to ensure consistency and clarity. Specifically, $u_*$ should be $\delta u$ as in equation 2, i.e., $\delta u := \mathcal{G}(u)$, where $\delta u$ is the solution of Eq. 2. The notation $u^*$ is the solution of nonlinear PDEs, namely, for an initial function $u_0$, assume the $n$-th iteration will approximate one solution, i.e., $(\mathcal{G} + \text{Id})^{(n)}(u_0) \approx u^*$.
>
> - 2.2 The definition of the Sobolev space can be found in Appendix B.1.  We rephrase the assumption here. Assuming that $X$ has a Schauder basis $\{b_n\}$, we define the canonical projection operator $P_n$ based on this basis. The operator $P_n$ projects any element $u \in X$ onto the finite-dimensional subspace spanned by the first $n$ basis elements $\{b_1, b_2, \ldots, b_n\}$. Specifically, for $u \in X$, $u = \sum_{k=0}^\infty \alpha_k b_k$, let
> $$
> P_n(u) = \sum_{k=0}^n \alpha_k b_k,
> $$
> where $\alpha_k$ are the coefficients in the expansion of $u$ with respect to the basis $\{b_n\}$.
> The canonical projection operator $P_n$ is a linear bounded operator on $X$. According to the properties of Schauder bases, these projections $P_n$ are uniformly bounded by some constant $C$. Furthermore, we assume, for any $u \in X$, $\epsilon > 0$, there exists an $n$ such that
> $$
> \|u - P_n u\|_{H^2(\Omega)} \le \epsilon, \quad \text{for all } u \in X.
> $$
> This ensures that $P_n$ effectively approximates elements of $X$ within the desired accuracy.
>
> - 2.3. $f(u)$ is a nonlinear function in $\mathbb{R}$ and $f'(u)$ is the derivative with respect to $u$, $u:\mathbb{R}^d \to \mathbb{R}$. For example, $f(u) := u^2$ and $f'(u)=2u$.
>
>  > DeepONet:
>
> - We understand the concern regarding the distinction between our method and standard DeepONet. The key difference is our integration of a Newton-based loss function, which is specifically designed to address the unique challenges posed by nonlinear PDEs. While existing convergence rate analyses typically focus on elliptic equations and linear advection–diffusion–reaction equations, our analysis in this paper is centered on the newly derived loss function tailored for nonlinear PDEs.
>
> ## Questions
>
> > Computational Efficiency:
>
> - For single linear Newton systems, our method is also significantly faster than traditional linear solvers. Furthermore, its primary advantage becomes more pronounced when solving multiple systems simultaneously. In this case, the learned operator greatly accelerates the solution process. This is because solving multiple nonlinear PDEs often involves computing many solutions. By generating solution data from various initial states, we can train the Newton operator on this diverse dataset. Efficiently sampling a wide range of initial states enhances the likelihood of converging to different solutions, which is crucial in complex solution landscapes with multiple basins of attraction.
>
> > Computational Time and Accuracy:
>
> - For the Gray-Scott model, we test both methods with the same set of initial conditions. Newton's method requires approximately 0.26 seconds to achieve an $L^2$ norm of the residual below $1 \times 10^{-4}$, while the neural operator only requires 0.00028 seconds to achieve an $L^2$ norm of the residual around $1 \times 10^{-1}$.
>
> > Approximation:
>
> - The approximation abilities discussed are based on specific assumptions and setups unique to our approach, particularly in handling multiple solutions. For Steps 3, 4, and 5, there is a proof for the generalization of DeepONet, which can be directly applied to other approximation proofs if the task is to use DeepONet to solve the problem. The proofs in Steps 1 and 2 aim to clarify the approximation in our setup and ensure that the operator we tend to approximate is well-defined.
>
> >Referencing:
>
> - Thanks for the careful reading. We cite Theorem 2.1 in the reference paper. In that paper, they consider function approximation, whereas our paper focuses on operator learning. If you consider the output of the operator learning, it should be a function, allowing us to use that theorem directly.
>
> > Clarifications:
> - The Argyris element, which satisfies $H^2$ approximation, will be utilized. The proof of the error associated with this element is shown in _S. Brenner, The Mathematical Theory of Finite Element Methods, Springer, 2008_. This choice is because this approximation includes second-order derivatives information in the approximation.
>
> > Theorem 2
> - $E$ is the expectation of sample points since we randomly choose sample points. The theorem holds for all $\theta$ as we consider the uniform error. Non-uniform generalization error remains an open question in learning theory, even for function learning.
>
> > Eq.~(25)
> - Eq.~(25) is not a typo; it is based on the norm embedding, which we will discuss further in the paper.

---

> > ### Comment · Reviewer_6H5E · 2024-08-12
> >
> > Thank you very much for the  response. However, the third point of weaknesses is not fully addressed. I understand the contribution of designing Newton-based loss function to address nonlinear pde. However,  the final task is to learn the operator solution to the linearized equations.   Therefore I would like to see the difference between the neural network part and the approximation and generalization analysis used in this paper and those of the existing DeepOnet. Having said that, I feel this paper is a valuable work in the field of operator learning and the broader scientific ML, and I maintain my original positive score as is.

---

> > > ### Author Response · Authors · 2024-08-12
> > >
> > > Thanks for the reviewer's positive response. Here we want to mention that although we use DeepONet to learn the operator form of linear PDEs, there are still some differences between our work and the standard approach.
> > >
> > > First of all, in our work, the loss function introduces the residual error from PDEs, whereas the standard DeepONet is purely a supervised learning method as described by Lu Lu, Pengzhan Jin, and George Em Karniadakis in DeepONet: Learning nonlinear operators for identifying differential equations based on the universal approximation theorem of operators (arXiv preprint arXiv:1910.03193, 2019).
> > >
> > > Secondly, in the proof of approximation, our input includes not only $ u $ itself but also the second derivative of $ u $. This requires that the encoding in our input part should be smoother. This is why we need an encoding like Argyris elements (Assumption 1 (iv)) to add the extra derivative information to achieve the approximation. The well-definition of the operator needs to be checked since this is not a smooth map from $ L^2 \to H^2 $ like in the standard DeepONet task, where regularity increases. In our case, we are dealing with $H^2 \to H^2$, which requires more assumptions and discussions to ensure the operator is well-defined. This is addressed in Steps 1 and 2 of the approximation part. Therefore, we need methods like Argyris elements for input and need to ensure the smoothness of $ f $ and the eigenvalue distribution of $ \mathcal{L} $.
> > >
> > > Lastly, regarding generalization error, we consider not only the generalization error for $ L^2 $-loss (Theorem 1) but also for $H^2 $ (Corollary 1). The generalization error for $H^2$ requires a similar proof framework but with more stringent smoothness assumptions for the operator and input space compared to the generalization error for $ L^2 $-loss (Theorem 1). The standard DeepONet only considers $L^2 $-loss and does not include this part.
> > >
> > > In summary, our paper uses DeepONet incorporating derivative information, unlike the standard DeepONet. We will discuss these differences and provide more details in the revised paper.

---

### Author Rebuttal · Authors · 2024-08-05

We thank all the reviewers for their careful reading and insightful comments. We would like to clarify our motivation and summarize the pipeline of our method as follows:

**Motivation**

The Newton Informed Neural Operator, focuses on efficiently approximating the iteration of Newton's method for a given state, specifically in the context of solving nonlinear PDEs with **Pattern Formation** (isolated multiple solutions). The primary approach of our method is to enhance the computational efficiency of finding multiple solutions by enabling extensive exploration of initial conditions. Let us briefly explain the motivation and methodology of our approach.

Newton's method for finding the roots of a function $ f(x) = 0 $ iteratively updates an initial guess $ x_0 $ using the formula:
$$
x_{k+1} = x_k - J_f(x_k)^{-1} f(x_k),
$$
where $ J_f(x_k) $ is the Jacobian matrix of $ f $ at $ x_k $. This method converges to a solution $ x^* $ if the initial guess $ x_0 $ is sufficiently close to $ x^* $.

In the context of nonlinear partial differential equations (PDEs), solving the nonlinear system often requires multiple Newton iterations, which can be computationally expensive, particularly when computing multiple solutions. Traditional numerical solvers for Newton's method can become a bottleneck due to their high computational cost. Our Newton Informed Neural Operator addresses this challenge by learning the operator defined within Newton's iterations, thereby speeding up the process and improving efficiency.

Our method approximates one iteration of Newton's method on a given state using a neural operator. The neural operator can perform these Newton's iterations orders of magnitude faster than using traditional solvers for Newton's iteration, as demonstrated in Table 1 of the manuscript. This significant speed-up allows for the practical exploration of a much larger space of initial conditions in a reasonable timeframe.

By efficiently sampling a wide variety of initial states, our method increases the likelihood of converging to different solutions of nonlinear PDEs, especially in cases where the solution landscape is complex and includes multiple basins of attraction. Consequently, while Newton's method alone does not inherently guarantee finding multiple solutions, the combination of rapid computation and extensive initial condition sampling enhances the chances of identifying multiple solutions.

**Pipeline of the method**

To illustrate the logic of our algorithm, consider the following steps:

1. **Initial Sampling**: Generate a diverse set of initial conditions $ \{x_0^i\} $ to explore multiple solutions.
2. **Neural Network Approximation**: Use the trained neural operator $ G_f $ to approximate the Newton step for each initial condition:
   $$
   x_{k+1}^i = x_k^i - G_f(f(x_k^i)),
   $$
   where the trained neural operator $ G_f $ approximates the action of $ J_f(x_k)^{-1} $.
3. **Iteration**: Repeat the neural operator-based Newton steps until convergence for each initial condition.
4. **Solution Identification**: Collect and analyze the converged solutions to identify distinct solutions of the PDE.

We will expand on this explanation in the revised manuscript to make the benefits and rationale of our approach more explicit. Thank you for highlighting this important aspect, which allows us to refine the presentation of our work.

---

### Decision · Program_Chairs · 2024-09-25

**Decision:**

Accept (poster)

**Comment:**

This paper:
* studies an interesting new setting (solving nonlinear PDE systems with multiple solutions)
* proposes an interesting new technique (Newton-informed neural operators)
* supports the technique with theoretical analysis as well as challenging experiments
* was reviewed positively by several experts in scientific machine learning.

Given the above points, this should have been a fairly clear case! However, the initial scores were low, and the reason might have been the presentation. The authors provided elaborate responses to the reviews which helped clarify several points. Overall, I feel that the presentation is a fixable issue, and overall the paper is of high quality.

For future revisions, I would encourage the authors to significantly revise Section 3.1 of the manuscript (it's far too terse), and explain very clearly (perhaps with a figure?) exactly what the Newton-informed neural operator is doing. I would also perhaps clarify the "multiple solutions" part up front a bit more clearly; really, the method speeds up the inner loop in Newton's method, enabling more rapid exploration of the solution space.